# Iron atom–cluster interactions increase activity and improve durability in Fe–N–C fuel cells

Xin Wan [1], Qingtao Liu [1], Jieyuan Liu [1], Shiyuan Liu [1], Xiaofang Liu[1], Lirong Zheng[2], Jiaxiang Shang[1], Ronghai Yu[1] & Jianglan Shui [1✉]

Simultaneously increasing the activity and stability of the single-atom active sites of M–N–C catalysts is critical but remains a great challenge. Here, we report an Fe–N–C catalyst with nitrogen-coordinated iron clusters and closely surrounding Fe–N$_4$ active sites for oxygen reduction reaction in acidic fuel cells. A strong electronic interaction is built between iron clusters and satellite Fe–N$_4$ due to unblocked electron transfer pathways and very short interacting distances. The iron clusters optimize the adsorption strength of oxygen reduction intermediates on Fe–N$_4$ and also shorten the bond amplitude of Fe–N$_4$ with incoherent vibrations. As a result, both the activity and stability of Fe–N$_4$ sites are increased by about 60% in terms of turnover frequency and demetalation resistance. This work shows the great potential of strong electronic interactions between multiphase metal species for improvements of single-atom catalysts.

[1] School of Materials Science and Engineering, Beihang University, 100191 Beijing, China. [2] Beijing Synchrotron Radiation Facility, Institute of High Energy Physics, Chinese Academy of Sciences, 100049 Beijing, China. ✉email: shuijianglan@buaa.edu.cn

Pyrolyzed metal-nitrogen-carbon (M–N–C) catalysts are highly efficient for many chemical reactions, including the oxygen reduction reaction (ORR) in proton exchange membrane fuel cells (PEMFC), and therefore regarded as promising low-cost alternatives to Pt/C catalyst[1–6]. The overall activity of M–N–C catalysts can be promoted either by maximizing the active site density (SD)[7–11], or by enhancing the turnover frequency (TOF) of a single site. The latter could be realized by the atomic level regulation of the geometric and electronic structures of the active sites, so as to optimize the adsorption/desorption of ORR intermediates[12–17]. However, so far, the ORR activity of M–N–C in acidic media is still significantly lower than that of Pt-based catalysts due to insufficient accessible active sites and less competitive TOF[8,11,18]. More importantly, the stability of M–N–C is far from satisfactory in real PEMFC[19–24]. The major causes of instability include the oxidation of carbon supports and the demetalation of M–N$_x$ active sites[25–28]. Anchoring active sites on highly graphitic carbons such as carbon nanotubes and graphene can improve the catalyst stability by enhancing corrosion resistance of the support[29,30]. However, such strategy often works at the cost of SD or TOF[31,32]. As for the anti-demetalation strategy, this is still rare. To develop methods that can break the activity–stability trade-off for M–N$_x$ species is essential for both theoretical investigation and practical applications of M–N–C catalysts.

Fe–N–C is the most active component among all M–N–C catalysts for ORR in acid. Fe–N–C catalysts may contain multi-scale metal phases from single atoms (SAs), atomic clusters (ACs) to nanoparticles (NPs) depending on the metal contents and the synthesis methods[33–35]. Recent studies have shown that the electronic interactions between SA active sites and metal NPs/ACs can enhance the activity of single-atom catalysts (SACs)[36–42]. Most of these synergies are demonstrated in alkaline media because these metal NPs/ACs are readily soluble in acids. It suggests that these metal NPs/ACs are weakly anchored (or bonded) on the carbon support, which may result in a very limited regulation effect on the electronic configuration of the SA sites. Fe NPs can be present in acids when they are encapsulated by a few layers of graphitic carbon ($L > 3$)[43,44]. However, theoretical calculations reveal that the electron penetration becomes very faint if the layer number is more than three, due to the quick drop of electron potential with the distance ($U_e = -ke^2/r$)[45]. It has been shown that if the ACs are chemically bound to the carbon support, they are acid resistant[46]. We therefore expect that acid-stable and closely adjacent Fe ACs and SAs should have much stronger electronic interactions than previously reported composite systems. It is also more worthy of expecting this strong interaction on the stability of Fe–N–C in acidic media and real PEMFC devices, which has not been explored either. Furthermore, it is still very challenging to predict the stability of Fe–N$_x$ active sites under PEMFC operating conditions by theoretical computational methods. Although attempts have been made by density functional theory (DFT) thermodynamic calculations, from the perspective of formation energy[20,47] or demetalation energy[19], these methods are oversimplified and cannot consider factors such as PEMFC temperature and voltage, which are known to have a large impact on catalyst stability[48]. Therefore, theoretical prediction methods for active site stability still need to be upgraded to incorporate practical operating conditions.

Herein, we synthesized N-anchored Fe ACs and satellite Fe–N$_4$ sites on two-dimensional porous carbon (Fe$_{SA}$/Fe$_{AC}$−2DNPC) as an efficient and stable ORR catalyst in acidic media. The introduction of Fe clusters is based on the utilization of protonated N-doped carbon substrate that has a moderate coordination strength to metal during the heat treatment, thus achieving balanced dispersion of Fe SAs and clusters on substrate. It is experimentally and theoretically demonstrated that Fe cluster can boost the activity of satellite Fe–N$_4$ site by introducing an OH ligand that reduces the ORR energy barrier. Molecular dynamics (MD) simulations are used for the stability prediction of Fe–N$_x$ at varying operating temperatures. A pinning effect of iron clusters is revealed, which shortens the amplitude of Fe–N bonds of satellite Fe–N$_4$ by incoherent vibrations of iron cluster and SAs. In this way, the demetalation of Fe–N$_4$ is reduced by 60%. In PEMFC, Fe$_{SA}$/Fe$_{AC}$−2DNPC exhibited an ultrahigh mass activity and promising long-term stability and durability, superior to the traditional Fe–N–C single-atom catalyst. These results demonstrate that the strong coupling of single atom and cluster is an effective strategy to improve the intrinsic activity and stability of single-atom active sites.

## Results

**Catalyst synthesis and morphology characterization.** Fe$_{SA}$/Fe$_{AC}$−2DNPC was synthesized via pyrolyzing a mixture of N-doped carbon quantum dots (CQD) and TPI (a Fe(II)-phenanthroline complex) using ice/silica dual templates. In brief, CQD was synthesized by acid-leaching half-carbonized zeolite imidazole frameworks (ZIF-8), which produced a protonated surface on CQD (Supplementary Figs. 1 and 2). Silica spheres with a diameter of ~100 nm were prepared by the Stöber method (Supplementary Fig. 3). The well-dispersed colloidal solution of CQD, SiO$_2$ spheres and TPI was freeze-dried to form a CQD/TPI@SiO$_2$ composite foam. Zeta potentials of these precursors are different as shown in Supplementary Fig. 4. During the freezing, negatively charged SiO$_2$ spheres were squeezed into a two-dimensional array at the interface of ice crystals, while positively charged CQD/TPI filled the voids of SiO$_2$ stacks, thereby forming a 2D inverse-opal porous structure. The freeze-dried foam was pyrolyzed to Fe$_{SA}$/Fe$_{AC}$−2DNPC after removing the silica. The pyrolysis temperature was optimized to 1000 °C to achieve the best activity and stability (Supplementary Figs. 5–7). The high temperature is crucial for the formation of optimal active sites and highly graphitic carbon support[49]. The catalyst was refluxed in a hot acid to remove soluble metal phases. As such, the remaining iron species should be robust to withstand the acidic environment.

The scanning electron microscopy (SEM) images (Fig. 1a, b) of Fe$_{SA}$/Fe$_{AC}$−2DNPC exhibit a micron-sized 2D nanosheet structure, constructed by 1–2 layers of open hollow spheres with an average diameter of ~80 nm. The transmission electron microscopy (TEM) image shows the interconnected ultrathin sphere walls without visible iron agglomerates (Fig. 1c). Atomic-resolution high-angle annular dark-field scanning TEM (HAADF-STEM) was performed to investigate the distribution of iron species at the atomic scale. As shown in Fig. 1d and Supplementary Fig. 8, the coexistence of Fe SAs and single-layer clusters is observed on carbon support. The magnified image in Fig. 1e more clearly shows that several iron atoms (red circles) closely surround a cluster (cyan circle) with a distance <0.5 nm, indicating the successful construction of Fe ACs and satellite SAs. The short inter-site distance allows rapid transfer of electrons from the ACs to the SAs. Graphene fringes are not observed around the cluster, indicating that the acid resistance of Fe ACs is not due to the protection of the graphene encapsulation. The elemental mapping further demonstrates the uniform distribution of the hybrid sites on nitrogen-doped carbon support (Fig. 1f). From HAADF-STEM images, we estimate that the average diameter of Fe ACs is 0.7 nm and the ratio of SA to AC is about 10:1 (Supplementary Fig. 9). We note that there is a fraction of SAs far away from the ACs, which should behave like regular single-atom active sites.

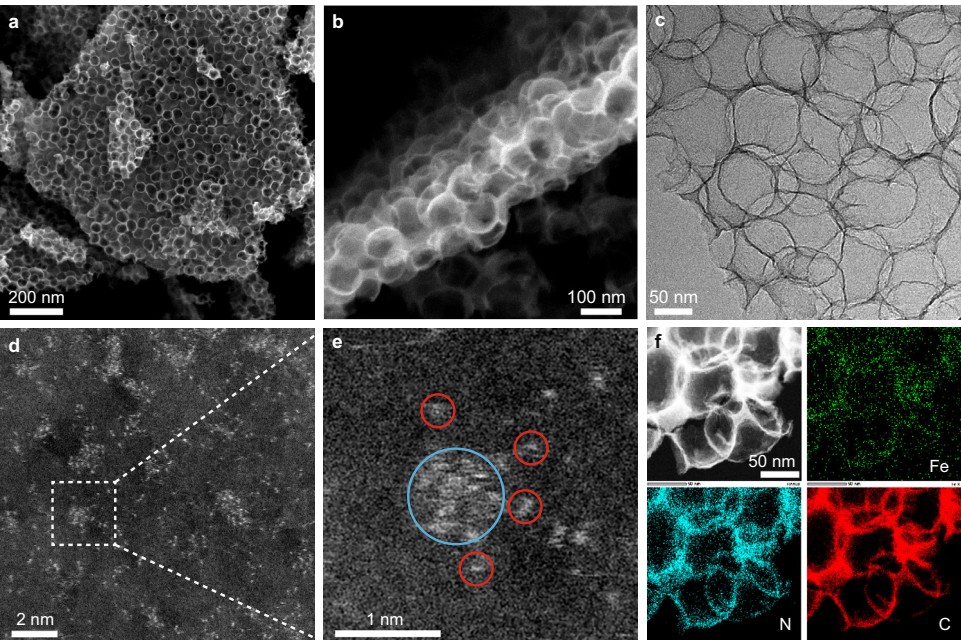

**Fig. 1 Morphology characterization of Fe$_{SA}$/Fe$_{AC}$−2DNPC. a, b** SEM images; **c** TEM image; **d, e** HAADF-STEM image with zoom-in image showing an iron cluster (cyan circle) and its satellite iron atoms (red circles); **f** HAADF-STEM image and corresponding element mappings.

**Active site structure analysis of Fe$_{SA}$/Fe$_{AC}$−2DNPC**. The X-ray photoelectron spectroscopy (XPS) N 1$s$ spectrum reveals the existence of pyridinic N (~398.4 eV), Fe–N bonding (~399.3 eV), graphitic N (~401.0 eV) and oxidized N (~403.1 eV), demonstrating the presence of Fe–N moieties in Fe$_{SA}$/Fe$_{AC}$−2DNPC (Fig. 2a, b and Supplementary Fig. 10)[50]. Notably, the Fe 2$p$ spectrum of Fe$_{SA}$/Fe$_{AC}$−2DNPC shows the positively charged iron species without obvious zero-valent iron (~706.7 eV), indicating that Fe atoms in the clusters are possibly coordinated by the substrate N/C atoms. The fine structure of the iron species was analyzed by X-ray absorption spectroscopy (XAS). Figure 2c shows the Fe K-edge X-ray absorption near-edge structure spectra of Fe$_{SA}$/Fe$_{AC}$−2DNPC and reference samples. The absorption threshold position of Fe$_{SA}$/Fe$_{AC}$−2DNPC is close to that of phthalocyanine (FePc) and far away from those of Fe foil and Fe$_2$O$_3$, implying that the chemical valence of iron is around +2 in Fe$_{SA}$/Fe$_{AC}$−2DNPC, consistent with the XPS results. The Fourier-transformed extended X-ray absorption fine structure (EXAFS) spectrum of Fe$_{SA}$/Fe$_{AC}$−2DNPC (Fig. 2d) shows a predominant peak at 1.5 Å in $R$ space, close to the Fe–N peak of FePc. A small peak corresponding to Fe–Fe path at around 2.3 Å is also observed. The FT-EXAFS spectrum was well fitted using backscattering paths of Fe–N/O and Fe–Fe (Fig. 2d, Supplementary Table 1). The coordination numbers of Fe–N/O and Fe–Fe were about 5.17 and 0.72, respectively. EXAFS wavelet transforms (WT) plot, a powerful method to distinguish the backscattering atoms, exhibits an intensity maximum at ~5 Å$^{-1}$ in $k$ space that was assigned to Fe–N/O, and a second intensity maximum at ~6.5 Å$^{-1}$ that could be ascribed to the Fe–Fe scattering (Fig. 2e). The negative shift of the $k$ value referring to the Fe–Fe bond of Fe foil (~8 Å$^{-1}$) may associate with the different coordination numbers between bulk Fe and Fe ACs[51,52]. Overall, the above characterizations prove that the Fe atoms exist as both mononuclear and multinuclear centers, and all are coordinated and stabilized by the support.

By varying the ratio of TPI in the precursor (5%, 15% and 30% of the CQD mass), the existing form of iron could be gradually tuned from SAs, to SAs/ACs, and then to SAs/NPs. The iron species on Fe$_{SA}$−2DNPC and Fe$_{SA}$/Fe$_{NP}$−2DNPC are characterized by X-ray

diffraction, HAADF-STEM and XAS in Supplementary Figs. 11–16. Fe$_{SA}$−2DNPC shows atomic iron only, while Fe$_{SA}$/Fe$_{NP}$−2DNPC has atomic iron and iron NPs encapsulated by graphitic layers. According to inductively coupled plasma optical emission spectrometry analysis, the overall iron contents are 0.59, 1.16, and 1.56 wt% for Fe$_{SA}$−2DNPC, Fe$_{SA}$/Fe$_{AC}$−2DNPC, and Fe$_{SA}$/Fe$_{NP}$−2DNPC, respectively. XPS analysis shows that iron was enriched on the catalyst surface (Supplementary Table 2). Fe$_{SA}$/Fe$_{AC}$−2DNPC has a high specific surface area of 995 m$^2$ g$^{-1}$, and is micropore-dominant as evidenced by the Type-I N$_2$ sorption isotherms with a sharp uptake at the low relative pressure ($P/P_0 < 0.015$) (Supplementary Fig. 17). Pore size distribution shows that micropores are mainly distributed at ~0.75 and ~1.30 nm, while mesopores are less pronounced. The high gas uptake at the high relative pressure ($P/P_0 > 0.9$) is indicative of the presence of a large number of macropores. Therefore, Fe$_{SA}$/Fe$_{AC}$−2DNPC possesses a pore structure composed of micropores and macropores, which serve as active site hosts and fast mass-transport channels, respectively. The other two catalysts have similar porosity and surface areas (Supplementary Fig. 17 and Supplementary Table 3). Therefore, the differences in the electrochemical performances, as discussed later, could be ascribed to the different forms of iron species on three Fe–N–C catalysts.

**Half-cell tests and quantitative analysis of active sites**. The ORR activity of synthesized catalysts was first evaluated by rotating ring disk electrode (RRDE) in O$_2$-saturated 0.5 M H$_2$SO$_4$ solution. Among three Fe-based catalysts, Fe$_{SA}$/Fe$_{AC}$−2DNPC shows the highest ORR activity in terms of half-wave potential ($E_{1/2}$) of 0.81 V vs. reversible hydrogen electrode (RHE) and Tafel slope of 54.5 mV dec$^{-1}$ (Fig. 3a and Supplementary Fig. 18). In the potential window of 0.2–0.8 V, the H$_2$O$_2$ yields are below 4% while the electron transfer numbers are above 3.9 for all catalysts (Fig. 3b), indicating four-electron ORR processes. To further understand the effect of the strong atom-cluster interaction on the intrinsic activity of Fe–N$_4$ site, the apparent activity of the catalyst was deconvoluted to SD and TOF by the in situ electrochemical method of nitrite adsorption and stripping, which is

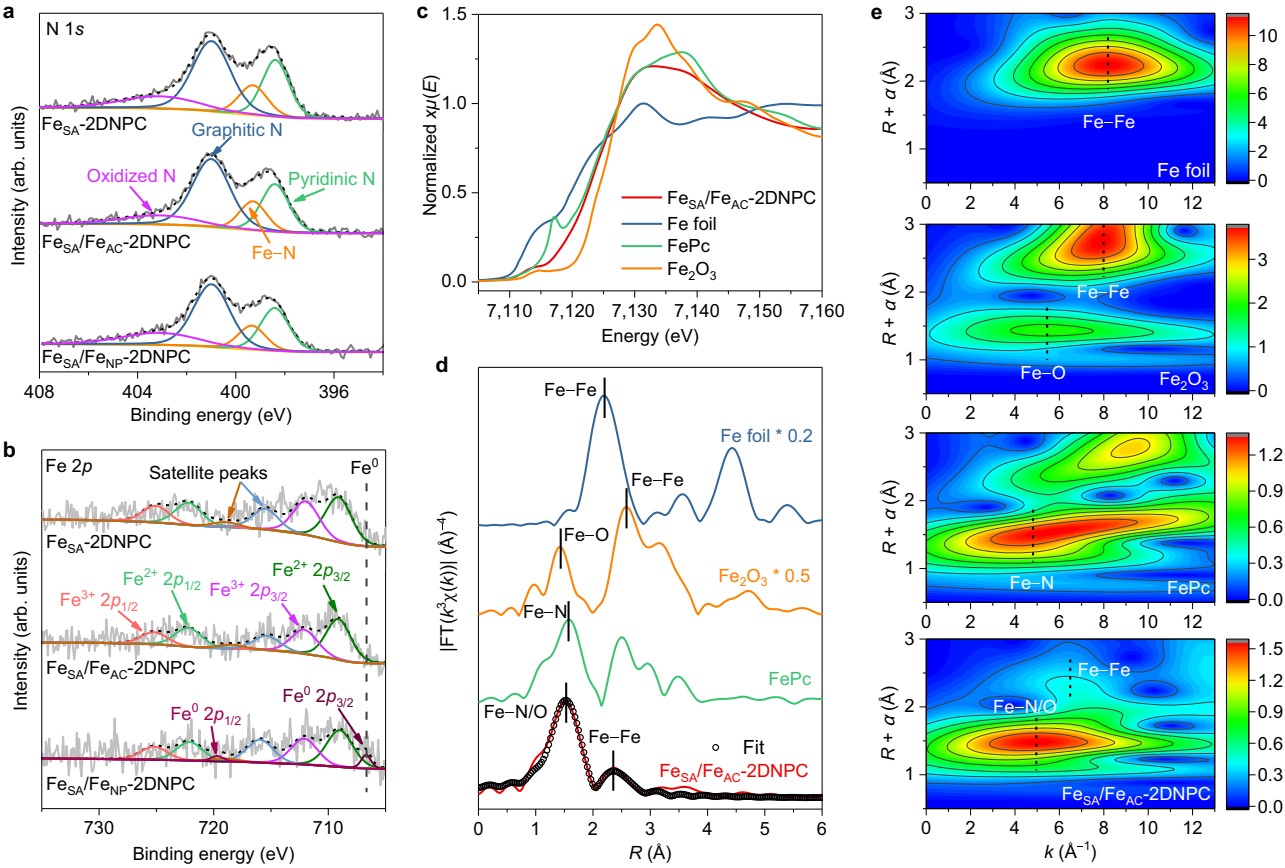

**Fig. 2 Active site structure analysis of Fe$_{SA}$/Fe$_{AC}$—2DNPC. a, b** High-resolution N 1$s$ (**a**) and Fe 2$p$ (**b**) XPS spectra of Fe$_{SA}$—2DNPC, Fe$_{SA}$/Fe$_{AC}$—2DNPC, Fe$_{SA}$/Fe$_{NP}$—2DNPC. **c** Normalized Fe K-edge XANES spectra of Fe$_{SA}$/Fe$_{AC}$—2DNPC and references of Fe foil, FePc and Fe$_2$O$_3$. **d, e** $k^3$-weighted Fourier transforms (**d**) and wavelet transforms (**e**) of the experimental EXAFS spectra of Fe$_{SA}$/Fe$_{AC}$—2DNPC and references of Fe foil, FePc and Fe$_2$O$_3$. FT-EXAFS fitting curve of Fe$_{SA}$/Fe$_{AC}$—2DNPC is present in (**d**). Source data are provided as a Source Data file.

only sensitive to the ORR active site of Fe–N$_4$ rather than Fe AC and NP (Supplementary Figs. 19–21 and Supplementary Table 4)[53]. The results are summarized in Fig. 3c. Although with the highest activity, Fe$_{SA}$/Fe$_{AC}$—2DNPC does not possess the highest SD among the three catalysts. Instead, a decrease in SD is observed with the emergence of Fe ACs and NPs, which consume considerable iron sources while contribute negligible activity[33]. The lower SD but higher apparent activity of Fe$_{SA}$/Fe$_{AC}$—2DNPC compared with Fe$_{SA}$—2DNPC points to a nearly 60% TOF enhancement (2.82 vs. 1.79 s$^{-1}$) of the Fe–N$_4$ site due to the incorporation of Fe cluster. The presence of Fe NPs also promotes the TOF of Fe–N$_4$ but to a small extent, suggesting that Fe ACs serve as a more powerful promoter to the activity of Fe–N$_4$ compared with the encapsulated NPs.

The stability of the developed catalysts was also evaluated in the half-cell as shown in Fig. 3d–f. After 10,000 potential cycles between 0.6 and 1.0 V in an O$_2$-purged 0.5 M H$_2$SO$_4$ at ambient temperature, Fe$_{SA}$/Fe$_{AC}$—2DNPC exhibits the best stability with an $E_{1/2}$ loss of only 15 mV, much smaller than those of Fe$_{SA}$—2DNPC (53 mV) and Fe$_{SA}$/Fe$_{NP}$—2DNPC (40 mV). Stability tests were also performed by chronoamperometry at 0.75 V for 20 h (Supplementary Fig. 22). We observed a 79% current density retention for Fe$_{SA}$/Fe$_{AC}$—2DNPC, outperforming Fe$_{SA}$—2DNPC (58%) and Fe$_{SA}$/Fe$_{NP}$—2DNPC (60 %) again. To evaluate the stability of the catalysts at a more practical temperature, we raised the temperature to 80 °C. Notably, after 5000 potential cycles, Fe$_{SA}$/Fe$_{AC}$—2DNPC still demonstrates extraordinarily high stability relative to the references, with a minimal $E_{1/2}$ loss of only 20 mV (Supplementary Fig. 23). Chronoamperometry tests at

80 °C and 0.75 V for 20 h also show that Fe$_{SA}$/Fe$_{AC}$—2DNPC has the best current density retention (Supplementary Fig. 24). All the above results demonstrate that the satellite Fe–N$_4$ sites around the Fe cluster have higher stability compared to the isolated Fe–N$_4$ sites.

Carbon corrosion and Fe–N$_4$ demetalation have been identified as the two most likely degradation mechanisms of Fe–N–C catalysts. Carbon corrosion is associated with the graphitization degree of carbon support. Figure 3g shows the Raman spectra of the catalysts, displaying two prominent peaks at ~1320 and 1588 cm$^{-1}$ assigned to the D band (crystal defects) and G band (in-plane stretching of sp$^2$ C) of carbon species, respectively. The approximately equal $I_D/I_G$ ratios indicate a similar degree of graphitization of the three catalysts, which is controlled by the pyrolysis temperature regardless of the iron content (Supplementary Fig. 7a). Therefore, the difference in the stability of the three catalysts is independent of the graphitic degree of the carbon supports. In addition, the ORR selectivity of the catalysts after potential cycles remains almost unchanged (see inset in Fig. 3d–f), implying the insignificant oxidation of the carbon support[25]. It is thus speculated that the enhanced stability of Fe$_{SA}$/Fe$_{AC}$—2DNPC is due to the strong electronic interaction between Fe–N$_4$ and Fe cluster, which lowers the tendency of the demetalation of Fe–N$_4$ sites. To prove this hypothesis, we quantified the demetalation of the catalysts after 5000 potential cycles. For Fe$_{SA}$—2DNPC, the amount of leached Fe was as high as 35.6%, whereas the Fe loss in Fe$_{SA}$/Fe$_{AC}$—2DNPC was greatly reduced to 15.5% (Fig. 3h and Supplementary Table 5). However, due to the complex composition of the catalysts with ACs or NPs,

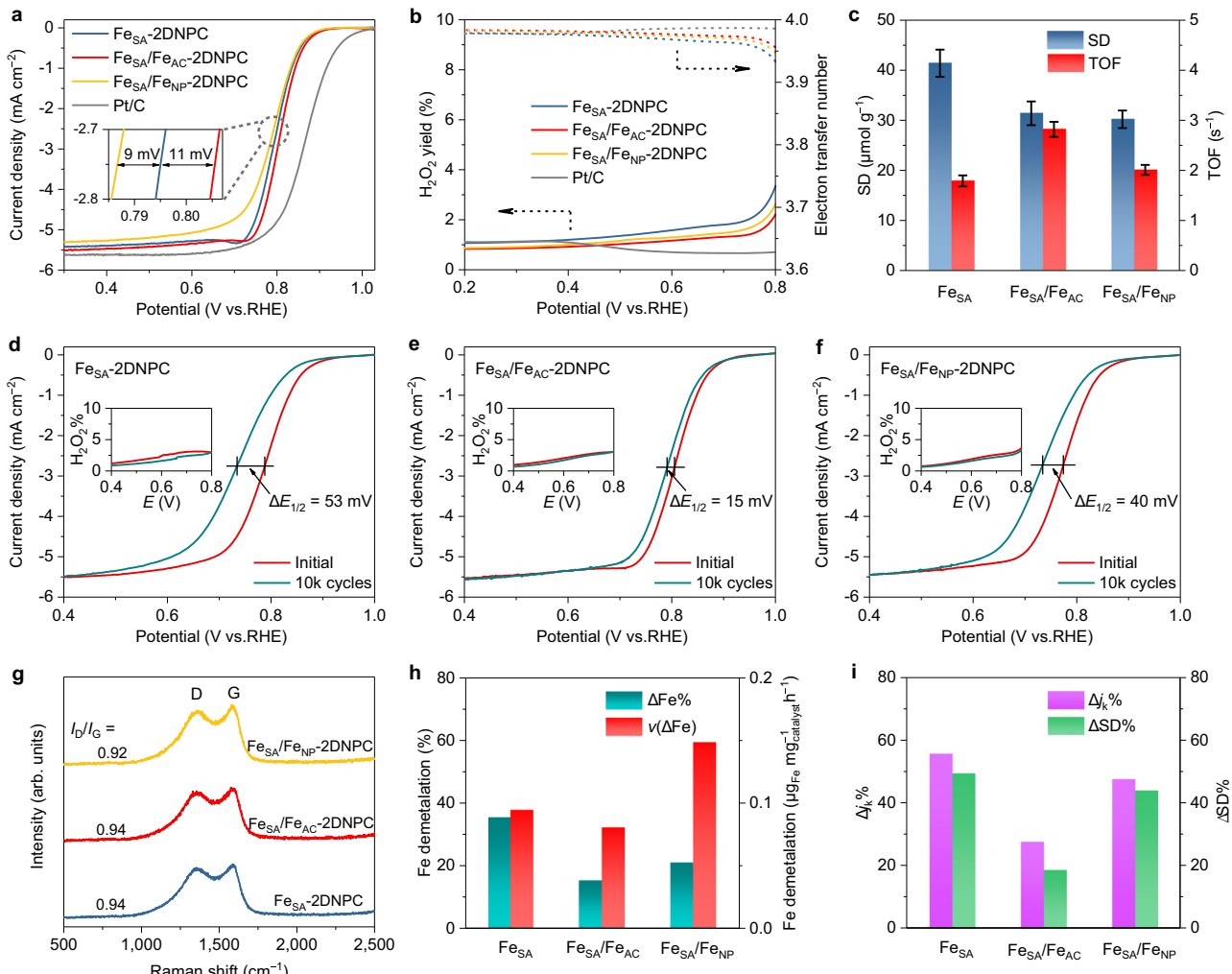

**Fig. 3 Half-cell tests and quantitative analysis of active sites. a** ORR polarization curves and (**b**) $H_2O_2$ yields and electron transfer numbers of $Fe_{SA}$−2DNPC, $Fe_{SA}/Fe_{AC}$−2DNPC, $Fe_{SA}/Fe_{NP}$−2DNPC and Pt/C in $O_2$-saturated 0.5 M $H_2SO_4$ (0.1 M $HClO_4$ for Pt/C). **c** SD and TOF of Fe-$N_4$ sites of the indicated Fe-N-C catalysts. Error bars correspond to the standard deviation of three-time measurements. **d**–**f** ORR polarization curves and $H_2O_2$ yields (inset) before and after 10,000 potential cycles (0.6-1.0 V vs. RHE) in $O_2$-purged 0.5 M $H_2SO_4$. **g** Raman spectra of the catalysts. **h** Results of metal leaching experiments. ΔFe%, relative amount of demetalation; $v(\Delta Fe)$, demetalation rate. **i** The changes of kinetic current density ($j_k$) and SD after the CV cycling. Source data are provided as a Source Data file.

it is hard to identify the types of leached Fe species and whether they are responsible for the performance decline. Again, we monitored the changes in Fe–$N_4$ SD and TOF (Fig. 3i, Supplementary Figs. 25–27 and Supplementary Table 5). After the stability test, $Fe_{SA}/Fe_{AC}$−2DNPC showed the highest retentions of the kinetic current density (72.3%) and SD (81.3%), while the other two catalysts lost nearly half of their active sites. It can be deduced that the presence of Fe clusters reduced the demetalation of Fe–$N_4$ by about 60% compared to isolated Fe–$N_4$. We note that TOF of $Fe_{SA}/Fe_{AC}$−2DNPC decreased slightly from 2.60 to 2.39 s$^{-1}$, which can be explained by either the preferential loss of the more active sites at the carbon edge[31] or the mild surface oxidation[25,54]. The HAADF-STEM image of the used $Fe_{SA}/Fe_{AC}$−2DNPC after 5000 potential cycles (Supplementary Fig. 28) shows the well-retained SA/AC hybrid sites, confirming its structural stability during the ORR process. Therefore, it can be said that the presence of Fe clusters hinders the demetalation of satellite Fe–$N_4$.

**PEMFC tests**. From a practical point of view, it is important to evaluate the performance of Fe–N–C catalysts in $H_2$–air PEMFC.

Figure 4a–c show the initial polarization and power density plots and the corresponding performance after 100/150 h of stability test. In good agreement with the half-cell results, $Fe_{SA}/Fe_{AC}$−2DNPC exhibits the highest peak power density ($P_{max}$) of 0.34 W cm$^{-2}$ in 1.0 bar $H_2$-air PEMFC, compared to 0.28 W cm$^{-2}$ of $Fe_{SA}$−2DNPC and 0.26 W cm$^{-2}$ of $Fe_{SA}/Fe_{NP}$−2DNPC. In the 150 h stability test at 0.5 V, $Fe_{SA}/Fe_{AC}$−2DNPC showed a slight drop in performance during the first 32 h, which was associated with the unstable active sites located at the edge of the carbon support or isolated from iron clusters; then followed by a relatively stable performance of about 0.37 A cm$^{-2}$ until the end of the test. Its performance at the 150th hour is almost identical to that at the 100th hour. Such a degradation pattern means that the catalyst has long-term stability, as confirmed by the stability test of another fuel cell (Supplementary Fig. 29). This stability performance is very promising compared with reported M–N–C catalysts (Supplementary Table 7). In contrast, $Fe_{SA}$−2DNPC and $Fe_{SA}/Fe_{NP}$−2DNPC show fast and constant declines throughout the tests. Catalyst stability was also evaluated by square wave voltage cycling, where continuous surface oxidation and reduction cycles accelerate catalyst degradation and can simulate vehicle operation (Fig. 4d, protocol according to US DOE). $Fe_{SA}/$

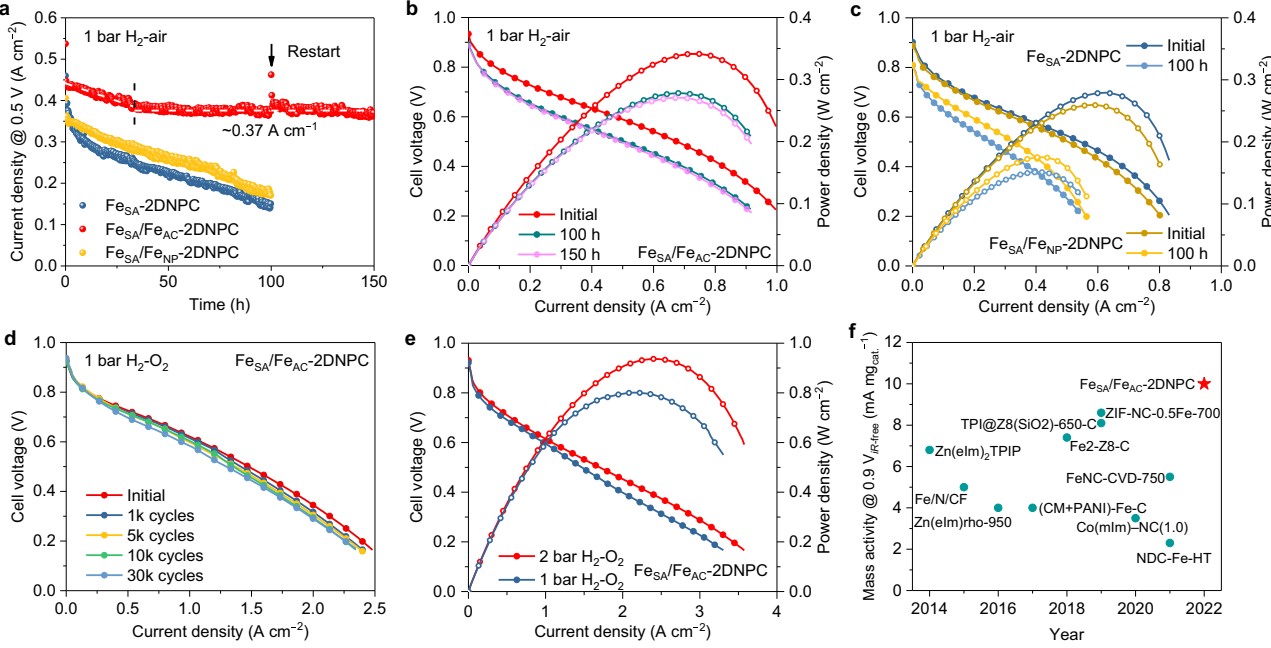

**Fig. 4 PEMFC tests. a–c** Polarization and power density curves of the indicated catalysts (**b**, **c**) before and after the stability test at a constant voltage of 0.5 V under 1 bar H$_2$-air (**a**). **d** Polarization curves of Fe$_{SA}$/Fe$_{AC}$−2DNPC (under 1 bar H$_2$-O$_2$) through 30,000 square-wave cycles between 0.6 and 0.95 V under H$_2$-N$_2$. **e** Polarization and power density curves of Fe$_{SA}$/Fe$_{AC}$−2DNPC under 1 and 2 bar H$_2$-O$_2$. Test conditions: cathode loading 1.5 mg$_{Fe-N-C}$ cm$^{-2}$ and, anode loading 0.2 mg$_{Pt}$ cm$^{-2}$, Nafion 211 membrane, 5 cm$^2$ electrode, 80 °C, 100% relative humidity (RH), flow rates of 300/600 ml min$^{-1}$ for H$_2$-air polarization, 100/100 ml min$^{-1}$ for H$_2$-air stability test, and 300/400 ml min$^{-1}$ for H$_2$-O$_2$ polarization. **f** Comparison of mass activity of the high-performing Pt-group-metal free catalysts in PEMFC under 1 bar H$_2$-O$_2$. The references to the data points are supplied in Supplementary Table 8. Source data are provided as a Source Data file.

Fe$_{AC}$−2DNPC showed 7.4% current loss at 0.6 V after 10,000 cycles and 17.2% after 30,000 cycles, outperforming previous reports of non-precious metal catalysts[55].

To gain better insight into the intrinsic activity of Fe$_{SA}$/Fe$_{AC}$−2DNPC, we performed fuel cell tests under 1.0 and 2.0 bar H$_2$–O$_2$ (Fig. 4e). $P_{max}$ reached 0.80 and 0.94 W cm$^{-2}$, respectively. The current density at 0.9 V$_{iR-free}$ (where $iR$-free indicates that the internal resistance is compensated for) is 15 mA cm$^{-2}$ under 1.0 bar H$_2$–O$_2$ (see Tafel plot in Supplementary Fig. 30). This value translates to a mass activity of 10 mA mg$_{cat.}^{-1}$ at 0.9 V$_{iR-free}$, outperforming most of the reported platinum-group-metal free catalysts (Fig. 4f and Supplementary Table 8). We note that the performance difference between 1 bar and 2 bar pressure is small even in the high current region. This phenomenon shows a small concentration effect due to the rapid mass transport in the interconnected porous structure of the catalyst[32,56]. We prepared a control structure of 2D flat-film (Supplementary Fig. 31a–c), using the same process as Fe$_{SA}$/Fe$_{AC}$−2DNPC but without the SiO$_2$ spheres. In RRDE test, the catalyst shows a considerable $E_{1/2}$ of 0.78 V but a small limiting current density of 4 mA cm$^{-2}$ (Supplementary Fig. 31d), indicating the difficult mass transport in the stacked 2D films. In fuel cell tests, this flat catalyst could hardly deliver a considerable current density (Supplementary Fig. 31e). This comparison clearly demonstrates the efficient mass transport capability of Fe$_{SA}$/Fe$_{AC}$−2DNPC.

**Theoretical analysis of the activity and stability of the hybrid active site.** To understand the role of Fe cluster in promoting the activity of satellite Fe–N$_4$, DFT calculations were performed. As shown in Fig. 5a, a model of Fe$_4$–N$_6$ with a closely adjacent Fe–N$_4$ site is built on graphene to mimic the hybrid active site of Fe$_{SA}$/Fe$_{AC}$−2DNPC. The distance of 4.97 Å between Fe SA and AC is consistent with the observation of HAADF-STEM. An isolated

Fe–N$_4$ is also constructed for comparison (Supplementary Fig. 32). We assumed a 4$e^-$ associative ORR pathway that proceeds through O$_2$*, OOH*, O*, and OH* (* denotes the adsorption site, Fig. 5b). The calculated energy diagrams at 1.23 V are presented in Fig. 5c. Consistent with our previous DFT calculations[47], the rate limiting step of ORR on Fe–N$_4$ is the formation of OH* (O* + H$^+$ + $e^-$ = OH*) with an energy barrier of 0.53 eV. When the iron cluster is introduced, Fe–N$_4$/Fe$_4$–N$_6$ shows strong adsorption to OH to the extent that a permanent OH ligand is grafted on Fe–N$_4$[32]. This OH ligand optimizes the binding strength of the other side of the Fe–N$_4$ site to the ORR intermediates, greatly reducing the limiting energy barrier to 0.35 eV. The Fe$_4$ in Fe–N$_4$-OH/Fe$_4$–N$_6$ is predicted with inferior activity (Supplementary Figs. 33 and 34), indicating the cluster mainly acts as an activity booster. Two variants of the cluster are further investigated using models of Fe$_{13}$–N$_6$ and Fe$_4$–C$_6$. The calculations show that the N-coordinated iron cluster has a more significant boosting effect on the adjacent Fe–N$_4$ sites than the C-coordinated iron cluster, while the number of Fe atoms in the cluster plays a less significant role (Supplementary Figs. 35 and 36).

Next, the stability of active sites was investigated by MD simulations from the perspective of the bond-length fluctuation, because the demetalation starts from the elongation and break of the Fe–N bond[57]. The radial distribution function (RDF) of Fe–N bond, $g_{Fe-N}(r)$, offers a direct measure of the frequency of appearance of N atom at a distance $r$ from the central Fe atom, thus can represent the distribution of Fe–N bond-length (Fig. 5d). If an Fe–N bond is long and widely distributed, it means that it is prone to fracture. At room temperature, $g_{Fe-N}(r)$ of Fe–N$_4$ shows three peaks in the range of 1.82–2.38 Å due to the thermal vibration of Fe–N bond. After the addition of the Fe cluster, the length distribution of Fe–N bonds narrows to 1.82–2.22 Å, indicating that the Fe–N bonds are more stable. In other words,

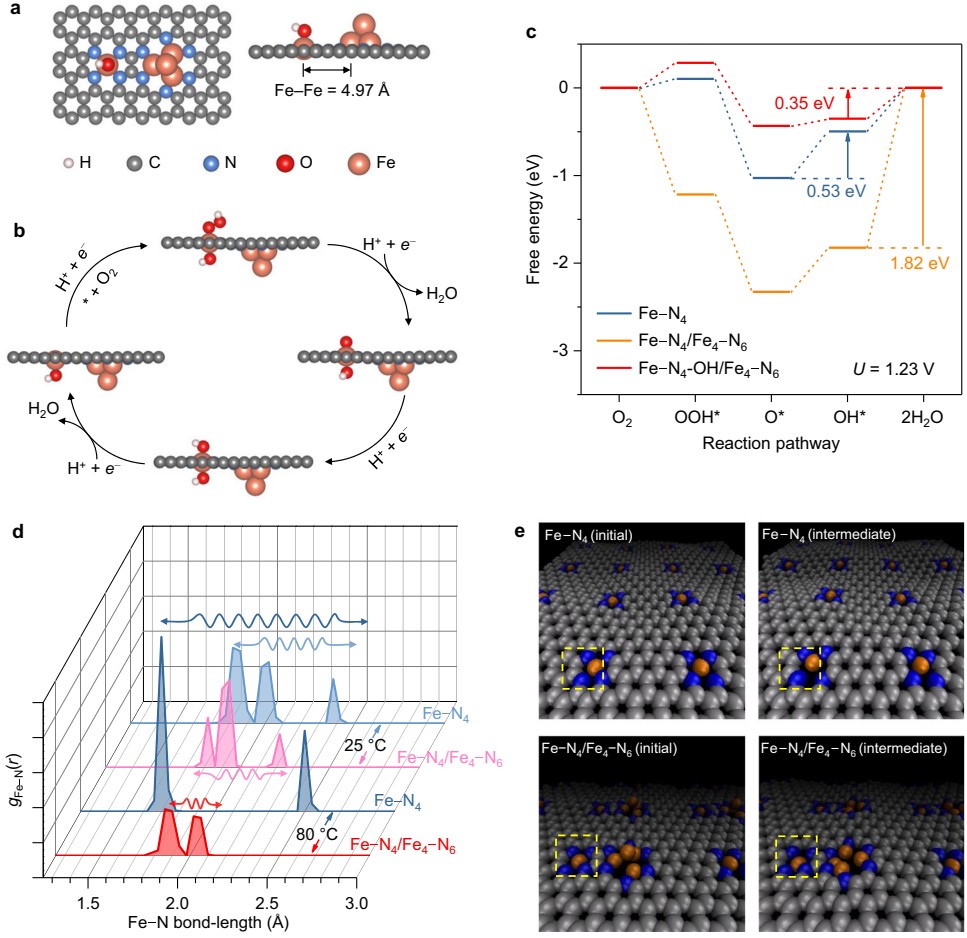

**Fig. 5 Theoretical analysis of the activity and stability of the hybrid active site. a** Model structure of $Fe-N_4/Fe_4-N_6$ used for theoretical calculation with a spontaneously formed OH ligand. **b** Schematic ORR process on the $Fe-N_4$ site of $Fe-N_4$-OH/$Fe_4-N_6$. **c** Free energy diagrams at 1.23 V for ORR over three types of active sites of $Fe-N_4$, $Fe-N_4/Fe_4-N_6$ and $Fe-N_4$-OH/$Fe_4-N_6$. **d** Fe–N radical distribution function profiles of the $Fe-N_4$ moiety in the models of bare $Fe-N_4$ and $Fe-N_4/Fe_4-N_6$ at 25 and 80 °C. Wavy arrows are used to indicate the amplitude of Fe–N bond-length fluctuation. **e** Snapshots of $Fe-N_4$ and $Fe-N_4/Fe_4-N_6$ obtained from MD simulations at 80 °C. The initial configuration (left) and an intermediate state (right) are provided to show the elongation of the Fe–N bond as marked by the yellow box. Source data are provided as a Source Data file.

the $Fe-N_4$ site in $Fe-N_4/Fe_4-N_6$ is more stable than an isolated $Fe-N_4$ site. At the PEMFC operating temperature of 80 °C, the Fe–N bond-length distribution increases to 1.71–2.50 Å in the isolated $Fe-N_4$, while it still maintains a narrow distribution of 1.86–2.06 Å in the $Fe-N_4$ of $Fe-N_4/Fe_4-N_6$. This explains why the degradation of $Fe_{SA}/Fe_{AC}-2DNPC$ was not substantially accelerated at the elevated temperature. The thermal vibrations of Fe–N bond of the $Fe-N_4$ and the $Fe-N_4/Fe_4-N_6$ can be visualized in Supplementary Movies 1–4. Snapshots obtained from MD simulations at 80 °C are shown in Fig. 5e for a quick comparison. Previous research has predicted that the vibration frequency of metal clusters increases with decreasing size down to diatomic molecules[58]. It is reasonable to assume that the incoherent vibration of Fe clusters and Fe SAs are responsible for the reduced amplitude of Fe–N bonds. Therefore, the iron clusters produce a pinning effect that suppresses the thermal vibrations of the satellite $Fe-N_4$ sites, thus reducing their tendency for demetalation.

## Discussion

In summary, a type of Fe–N–C catalyst has been synthesized with Fe cluster and satellite $Fe-N_4$ coupling active sites on carbon support. Despite complete exposure, the N-anchored Fe clusters are acid-stable. The iron cluster introduces an OH ligand on the

satellite $Fe-N_4$, thus lowering the ORR energy barrier and increasing the intrinsic activity of the $Fe-N_4$ site by 60%. Moreover, the method of MD simulation was introduced to observe the vibration and amplitude of the chemical bonds of $Fe-N_4$ active site at varying working temperatures of PEMFC, and revealed the stability-enhancing mechanism of the $Fe_{SA}/Fe_{AC}$ active site, namely the pinning effect. The incoherent vibrations of Fe clusters and Fe SAs suppress the amplitude of Fe–N bonds, resulting in a 60% reduction in the demetalation of $Fe-N_4$ sites as well. In the PEMFC device, $Fe_{SA}/Fe_{AC}-2DNPC$ achieves an ultrahigh mass activity of 10 mA $mg_{cat.}^{-1}$ at 0.9 $V_{iR\text{-free}}$ under 1 bar $H_2-O_2$, and significantly enhanced stability compared with the single-atom catalyst ($Fe_{SA}-2DNPC$) and nanoparticle/single-atom catalyst ($Fe_{SA}/Fe_{NP}-2DNPC$). The synthetic method, active site structure, and performance-enhancing mechanisms can be extended to other single-atom catalyst systems.

## Methods

**Chemicals**. Zinc nitrate hexahydrate ($Zn(NO_3)_2·6H_2O$, 99.99%), 2-methylimidazole (98%) and nitric acid ($HNO_3$, 65–68%) were purchased from Aladdin. Perchloric acid ($HClO_4$, 70–72%), sulfuric acid ($H_2SO_4$, A.R.), hydrochloric acid (HCl, 36–38%), acetic acid ($CH_3COOH$, 99.5%), methanol ($CH_3OH$, A.R.), ethanol ($C_2H_5OH$, A.R.), sodium nitrite ($NaNO_2$, A.R.) and sodium acetate anhydrous ($CH_3COONa$, A.R.) were purchased from Beijing Chemical Works. Tetraethyl orthosilicate (TEOS, $C_8H_{20}O_4Si$, 99%) was obtained from Innochem. Ferrous acetate ($Fe(C_2H_3O_2)_2$, 95%) was purchased from J&K Chemicals.

1,10-phenanthroline and sodium hydroxide (NaOH, 95%) were obtained from Macklin. Commercial Pt/C (40 wt%) was obtained from BASF. Nafion alcohol (5 wt %, D520) was obtained from Aldrich. Nafion 211 membrane was obtained from DuPont. All chemicals were used without further purification. All aqueous solutions were prepared using deionized (DI) water with a resistivity of 18.2 MΩ.

**Synthesis of CQD**. First, ZIF-8 NPs were prepared by quickly pouring a 200 ml methanol solution containing 5.88 g $Zn(NO_3)_2 \cdot 6H_2O$ into another 200 ml methanol solution of 2-methylimidazole (6.48 g) while stirring. The mixture was stirred for 5 h. The white precipitates were centrifuged, washed with methanol three times and dried in a vacuum at 60 °C. The obtained ZIF-8 powder was grinded and then underwent a heat treatment at 535 °C for 8 h under argon atmosphere in a tube furnace. The obtained brown powder was dispersed in 20 ml DI water and then etched to CQD by adding 5 ml concentrated hydrochloric acid (36– 38%) under ultrasonic vibration. Then the suspension was purified by membrane dialysis (molecular weight cut off 8000–14,000 Da) against ultrapure water for 2 days to obtain the CQD colloidal solution[59]. The concentration of the CQD colloidal solution was determined by a dry weighing method. A 100-μl aliquot of CQD colloidal solution was pipetted onto microscope slides and dried on a hot plate, and then the remaining solid was weighed. The yield of CQD was about 25 wt% from the raw ZIF-8 precursor.

**Synthesis of SiO₂ spheres**. Silica spheres were prepared by the Stöber method[60]. Typically, 100 ml ethanol, 6 ml DI water, 6 ml ammonium hydroxide and 3 ml TEOS were mixed and stirred for 5 h. Then the resulting suspension was centrifuged, washed with ethanol and water, and dried in a vacuum at 60 °C to obtain $SiO_2$ spheres with a diameter of about 100 nm.

**Synthesis of the catalysts**. In a typical procedure, a colloidal solution of CQD (3 mg ml⁻¹), SiO₂ sphere (9 mg ml⁻¹) and TPI (0.45 mg ml⁻¹, prepared by dissolving ferrous acetate and 1,10-phenanthroline with the molar ratio of 1:3 in water) was prepared and freeze-dried to form a CQD/TPI@SiO₂ foam. The foam was firstly pyrolyzed at 800 °C for 1 h under argon atmosphere in a tube furnace. The carbonized foam was ground for 20 min in a mortar and then dispersed in NaOH solution (3 mol l⁻¹) at 50 °C for 10 h to etch the silica spheres. The dispersion was separated by filtration and washed thoroughly and dried in a vacuum at 60 °C. The obtained black powder was subjected to a second pyrolysis in the tube furnace at 1000 °C for 1 h under argon atmosphere. The product was subsequently refluxed in 0.5 M $H_2SO_4$ solution at 80 °C for 5 h to remove any possible unstable metal species. The final catalyst $Fe_{SA}/Fe_{AC}-2DNPC$ was collected by filtration and washed thoroughly and dried in vacuum at 60 °C. The yield of the catalyst was about 20 wt% based on CQD.

$Fe_{SA}-2DNPC$ and $Fe_{SA}/Fe_{NP}-2DNPC$ were prepared by the same procedure but changing the concentration of TPI to 0.15 and 0.9 mg ml⁻¹, respectively. As another control sample, 2D-Fe-N-C was prepared without the use of the silica sphere template and the silica etching process, while other synthesis steps were identical to the synthesis of $Fe_{SA}/Fe_{AC}-2DNPC$.

**Characterizations**. The SEM was performed with JEOL JSM-7500. The TEM was performed with JEOL JEM-2100F with an electron acceleration energy of 200 kV. The images of single iron atoms and elemental mapping were obtained by a HAADF-STEM (JEOL JEM-ARM200F) operated at 200 kV. XRD patterns were recorded on Rigaku D/max 2500 with Cu Kα irradiation. XPS measurements were performed on Thermo ESCALAB 250Xi using Al Kα irradiation. XPS data analysis comprised a Shirley background subtraction and a least-square fitting procedure of the spectra using XPSPEAK software. Raman spectra were recorded on Renishaw inVia Raman microscope (λ = 514 nm). $N_2$ sorption isotherms were measured by the SSA-7000 system (Beijing Builder) at 77.3 K and the porosity parameters were analyzed using the software QuadraWin (version 6.0). The specific surface area was obtained using the Brunauer–Emmett–Teller (BET) method. The pore size distribution was determined using quenched solid DFT model for slit shaped and cylindrical pores[61]. The external surface area is defined as non-micropore area, which was obtained by subtracting the micropore surface area (calculated through t-plot method) from the BET surface area. The iron concentration measurements were conducted on the Optima-7000DV ICP-OES. A certain amount of catalyst was placed in an alundum boat and burned to iron oxide in a muffle furnace at 800 °C for 1 h. The obtained iron oxide was dissolved by aqua regia heated in an oil bath at 80 °C until it was clear, and then diluted to the ppm range for the ICP-OES measurements. The zeta potentials of the samples were determined using a Zetasizer Nano ZS90. The samples were dispersed by sonication in water with a concentration of 10 mg l⁻¹ and the dispersion was used for measurements without the pH adjustment. XAS was performed at room temperature on the 1W1B beamline at BSRF (Beijing Synchrotron Radiation Facility). The catalysts (~15 mg) were pelletized as disks of 8 mm diameter using paraffin as a binder, while the iron phthalocyanine was mixed with BN powder with a ratio of 1:6. Fluorescence-mode Fe K-edge X-ray absorption spectra were collected for all samples over a range of $6915 - 7891$ eV, where a 100% Ar filled Lytle ion-chamber detector with Mn X-ray filters and soller slits were used. The monochromator energy was calibrated using a Fe foil. The XAFS data were analyzed using IFEFFIT[62]. The XAFS raw data were

background subtracted, normalized and Fourier transformed by standard procedures within the ATHENA program[63]. Least-squares curve fitting analysis of the EXAFS $χ(k)$ data was carried out using the ARTEMIS program[63]. All fits were performed in the $R$ space with $k$-weight of 3. The amplitude reduction factor ($S_0^2$) was determined from the Fe foil and held constant for the analysis of the samples. The EXAFS $R$-factor, which indicates the percentage misfit of the theory to the data, was used to evaluate the goodness of the fitting.

**RRDE tests**. The ORR activity was measured in acid medium on a glassy carbon RRDE (5.61 mm of disk outer diameter, Pine Research Instrumentation, USA) with an electrochemical workstation (CHI 760E, CH Instruments). The reference electrode was a calibrated saturated calomel electrode and the counter electrode was a graphite rod. All the potentials reported in this work were calibrated to the RHE and not iR-compensated. Catalyst inks were prepared by dispersing 1 mg of catalyst in 200 μl Nafion solution (1 mg ml⁻¹), which was prepared by mixing 215 μl Nafion alcohol (5 wt%, Aldrich), 4.3 ml DI water and 5.485 ml isopropanol, with 30 min sonication to get a uniform suspension. Before the test, the glassy carbon electrode was polished and rinsed with DI water. Two aliquots of 10 μl of the catalyst ink were successively pipetted onto the glassy carbon and dried in air, resulting in a catalyst loading of around 400 μg cm⁻². A Pt/C (40 wt% of Pt, BASF) catalyst with a loading of 80 μg_{Pt} cm⁻² was used as a reference (only one aliquot of 10 μl of the catalyst ink was deposited on the glassy carbon).

ORR activity was recorded at room temperature (~25 °C) at a rotation rate of 1600 rpm. The electrolyte was 0.5 M $H_2SO_4$ for Fe–N–C catalysts, and 0.1 M $HClO_4$ for Pt/C. The electrolyte was purged by any specific gas for at least 30 min before the tests and the gas flow was maintained during the experiments. The electrolyte was purged with $O_2$ first, followed by linear sweep voltammetry (LSV) at a scan rate of 10 mV s⁻¹ for ORR activity tests. Afterward, the LSV curve at the same scan rate was collected to obtain capacitive background in the Ar-saturated electrolyte. The oxygen reduction currents were obtained by subtracting the background currents from the original LSV measured in the $O_2$-saturated electrolyte. The peroxide yields ($H_2O_2$%) were calculated from the ring current ($I_r$) and the disk current ($I_d$) using the equation [Eqs. 1 and 2]:

$$H_2O_2\% = 200 \times I_r / (I_r + NI_d) \tag{1}$$

The electron transfer number ($n$) in acid was calculated by the equation:

$$n = 4I_d / (I_d + I_r/N) \tag{2}$$

where $N = 0.37$ is the current collection efficiency of the Pt ring (ring potential = 1.27 V vs. RHE).

The accelerated stability tests (AST) were performed by potential cycling from 0.6 to 1.0 V vs. RHE in $O_2$-purged 0.5 M $H_2SO_4$ at a scan rate of 50 mV s⁻¹ and a rotation rate of 300 rpm. Chronoamperometry tests at 0.75 V vs. RHE at a rotation rate of 300 rpm were also performed to study the catalyst stability.

**Quantification of the active sites**. The active SD and TOF were obtained according to the method presented by Kucernak et al[53]. Briefly, the catalyst inks were prepared by dispersing 1 mg of catalyst in 200 μl Nafion solution (1 mg ml⁻¹). An aliquot of 6 μl of the catalyst ink was deposited on the glassy carbon (rotating disk electrode, diameter of 5 mm), resulting in a catalyst loading of around 150 μg cm⁻². Then extensive cycling in pH 5.2 acetate buffer alternatively in $O_2$ and $N_2$ was performed to obtain non-changing CV curves in $N_2$. Then the catalyst was poisoned by $NaNO_2$. ORR performance was recorded before, during and after the nitrite absorption. Nitrite stripping was conducted in the region of 0.35 to −0.35 V vs. RHE. The excess in cathodic charge ($Q_{strip}$) was proportional to the active SD, and the TOF was calculated by dividing the difference of kinetic current before and after nitrite absorption by SD [Eqs. 3 and 4]:

$$SD(mol\ g^{-1}) = \frac{Q_{strip}(C\ g^{-1})}{n_{strip}F(C\ mol^{-1})} \tag{3}$$

$$TOF(s^{-1}) = \frac{n_{strip}\triangle j_k(mA\ cm^{-2})}{Q_{strip}(C\ g^{-1})L_C(mg\ cm^{-2})} \tag{4}$$

where $n_{strip}$ (= 5) is the number of electrons associated with the reduction of one nitrite per site, $j_k = \frac{j_{lim} \times j}{j_{lim} - j}$ is the kinetic current density, $L_C$ is the catalyst loading (0.15 mg cm⁻²).

For the comparison of SD and TOF before and after stability test, the AST was performed in 0.5 M $H_2SO_4$ while SD was determined in pH 5.2 acetate buffer. We note that nitrite ions may be reduced by the trace amount of metallic iron (if any) in the catalysts[64]. However, the major products are ammonium and nitrogen gas, which should not interfere with adsorption and subsequent stripping of nitrite on the active Fe–$N_x$ sites.

**Demetalation experiments**. 2 mg of catalyst was mixed with 12 μl Nafion alcohol solution (5 wt%, Aldrich), 38 μl $H_2O$ and 50 μl isopropanol and sonicated for 30 min. 5 aliquots of 10 μl of the catalyst ink were successively deposited on the glassy carbon of rotating disk electrode (diameter of 5 mm), achieving a total catalyst loading of 1 mg. The electrode was subjected to 5000 potential cycling from

0.6 to 1.0 V vs. RHE in $O_2$-purged 0.5 M $H_2SO_4$ at a scan rate of 50 mV s$^{-1}$ and a rotation rate of 300 rpm at 25 °C. The post-testing catalyst was carefully transferred to an alundum boat with no remaining with the assistance of isopropanol. The alundum boat with the collected post-testing catalyst was heated in a muffle furnace at 800 °C for 1 h to convert the catalyst to iron oxide. The obtained iron oxide was dissolved in aqua regia at 80 °C and diluted to the ppm range for the ICP-OES measurements.

**PEMFC tests**. The Fe–N–C catalyst (~9 mg) was mixed with Nafion alcohol solution (5 wt%, Aldrich), DI water (200 mg) and isopropanol (400 mg) to prepare the catalyst ink. The Nafion-to-catalyst ratio (NCR) was 1.5. The ink was subjected to sonication for 10 min and stirring for 10 h to make a uniform suspension. The well-dispersed ink was brushed on a piece of carbon paper (5 cm$^2$, GDS2240, Ballard), followed by drying in vacuum at 80 °C for 2 h. As for anode, Pt/C (40 wt% of Pt, BASF) was used with a loading of ~0.2 mg$_{Pt}$ cm$^{-2}$. The prepared cathode and anode were pressed onto the two sides of a Nafion 211 membrane (DuPont) at 130 °C for 90 s under a pressure of 1.5 MPa to obtain the MEA. The performance of MEA was measured by a fuel cell test station (Scribner 850e) with UHP-grade $H_2$ and $O_2$ or air at 80 °C, 100%RH. The flow rates were 0.3 l min$^{-1}$ for $H_2$, 0.4 l min$^{-1}$ for $O_2$ and 0.5 l min$^{-1}$ for air. The pressure conditions for each fuel cell data were specified in the figure captions. For example, 1 bar $H_2$–$O_2$ means that the absolute pressure for $H_2$ and $O_2$ is both 1 bar, which is achieved by applying 0.5 bar backpressure. Polarization curves were recorded by scanning the current density with an increment of 20 mA cm$^{-2}$ and the system is allowed to equilibrate by 3 s at each current step before a data point is recorded. The $iR$-compensated cell voltage was used for the Tafel pot in Supplementary Fig. 30. The cell internal resistance was provided by the test station. When performing stability test at the constant voltage mode, the flow rate was switched to 0.1 l min$^{-1}$ for any gas. Fuel cell durability was assessed followed the US Department of Energy (DOE) protocols simulating automotive drive cycles using a voltage square wave (steps between 0.6 V (3 s) and 0.95 V (3 s) with rise time of ~0.5 s), in which successive cycles of surface oxidation and reduction will accelerate the catalyst degradation. During the voltage cycles, the cathode and anode were purged with 100%RH $H_2$ and $N_2$ at 80 °C, respectively. The flow rates were 0.2 l min$^{-1}$ for $H_2$ and 0.075 l min$^{-1}$ for $N_2$. Polarization curves were recorded under 1 bar $H_2$–$O_2$ before the durability test and after 0, 1k, 5k, 10k, and 30k cycles. Each test was repeated several times to obtain a relatively stable polarization curve for performance comparison.

**Computational methods**. DFT calculations were performed using Vienna Ab Initio Simulation Package. The interactions between valence electrons and ion cores were modeled by projector augmented wave based potentials. The generalized gradient approximation, as parameterized by Perdew-Burke-Ernzerhof was used to describe the electron exchange and correlation energy. A plane-wave kinetic energy cut off of 500 eV was adopted after a series of tests for all the calculated models. The Brillouin zone was sampled with a Monkhorst-Pack $3 \times 3 \times 1$ k-point grid. Geometries were optimized until the force was converged to 0.01 eV/Å. A large vacuum slab of 25 Å was inserted in the z direction for surface isolation to avoid the interaction between two neighboring surfaces for all the calculations.

The ORR in an acid electrolyte takes place through the following elementary steps [Eqs. 5–9][65]:

$$O_2(g) + H^+ + e^- + * \rightarrow OOH^* \quad (5)$$

$$OOH^* + H^+ + e^- \rightarrow O^* + H_2O(l) \quad (6)$$

$$O^* + H^+ + e^- \rightarrow OH^* \quad (7)$$

$$OH^* + H^+ + e^- \rightarrow H_2O(l) + * \quad (8)$$

where * stands for the active site on the catalyst, OOH*, O*, and OH* refer to adsorbed intermediates, (l) and (g) refer to liquid and gas phases, respectively.

The change of Gibbs free energy for each elementary step was calculated as follows:

$$\Delta G = \Delta E + \Delta ZPE - T\Delta S + \Delta G_U + \Delta G_{pH} \quad (9)$$

where $\Delta E$ refers to the reaction energy obtained from DFT calculations, $\Delta ZPE$ and $\Delta S$ are the change in zero-point energy and entropy, respectively. The effect of a bias on all the states involving an electron in the electrode is taken into account by shifting the electron energy of the corresponding state by $\Delta G_U = -eU$, when an electron is transferred. $\Delta G_{pH}$ is the free energy correction of the $H^+$, and is calculated by the equation: $\Delta G = k_B T \times \ln10 \times pH$, where $k_B$ is the Boltzmann constant and $T$ is the temperature. pH = 0 was taken for ORR in an acidic electrolyte. For $H_2O$ in the liquid phase, the gas phase of $H_2O$ at 0.035 bar, which is the equilibrium pressure of $H_2O$ at 298.15 K, was used as the reference system. The free energy of $O_2$ was obtained from the reaction $O_2 + 2H_2 \rightarrow 2H_2O$ due to the inaccuracy of DFT in estimating the cohesive energy of $O_2$. The overpotential of a catalyst for ORR is determined by [Eq. 10]:

$$\eta = \Delta G_{max}/e + 1.23 \text{ V} \quad (10)$$

where $\Delta G_{max}$ is the maximum Gibbs free energy difference between each two successive reaction steps at $U = 0$ V.

MD simulations were carried out in the Large-scale Atomic/Molecular Massively Parallel Simulator code[66]. The energy of the Fe–$N_4$ and Fe–$N_4$/Fe$_4$–$N_6$ system was minimized using the conjugate gradient algorithm prior to MD simulations. Periodic boundary condition was applied in all directions. The mixed interaction is defined via the hybrid command combining Tersoff [67], MEAM[68], and Lennard-Jones (LJ)[69] potential. Tersoff and MEAM potential were used for C–N, and Fe–N, Fe–Fe bonding interactions, respectively. While the interaction between Fe and C is modeled using the LJ potential. The time step was set up as 0.001 ps in the constant-pressure, constant-temperature ensemble (NPT). The Nosé−Hoover chain thermostat was applied for temperature control. Visual Molecular Dynamics (VMD) software[70] is used for 2D visualization of the atomic configurations.

## Data availability

The data supporting this study are available within the paper and the Supplementary Information. Source data are provided with this paper.

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

## Acknowledgements

This work was supported by Natural Science Foundation of Beijing Municipality (Z200012), National Natural Science Foundation of China (21975010, U21A20328) and the Academic Excellence Foundation of BUAA for PhD Students.

## Author contributions

J.Shu. and X.W. conceived and designed the research. X.W. conducted the synthesis, electrochemical measurements, and characterizations. Q.L., J.L., S.L., and J.Sha.

performed the theoretical calculations. L.Z. conducted XAS measurements. X.W., X.L., and J.Shu. co-wrote the paper. The project was supervised by R.Y. and J.Shu.

## Competing interests

The authors declare no competing interests.
