## [Peer Review File · Nature Communications]

Title: Iron atom-cluster interactions increase activity and improve durability in Fe-N-C fuel cellsREVIEWER COMMENTS

Reviewer #1 (Remarks to the Author):

In the manuscript by Wan et al., the authors report the construction of Fe-N-C electrocatalyst with single-atom/cluster hybrid active sites for high-performance fuel cells. The iron single atoms and clusters are stabilized by N/C on carbon support and closely adjacent, thus creating a strong interaction. It is interesting that this strong interaction simultaneously increases the activity and stability of the single-atom active sites. In recent years, stability challenge has become a central issue for the development of M-N-C fuel cell catalysts, while the mitigation strategies are still limited. This work is the first to show that the synergy between single-atom/cluster could improve the fuel cell stability. The authors also performed molecular dynamics simulations to investigate the mechanism for the stability enhancement, and proposed a pinning effect of iron clusters, which can shorten the amplitude of Fe-N bonds of the adjacent Fe-N₄ active sites and thus improve their stability.

The presented data are well organized and the editing quality of the manuscript is acceptable. Due to the promising results and the apparent scientific quality, I would recommend the manuscript for publication in Nature Communications after solving the following issues.

1. It is often very challenging to well define the structure of metal clusters. In DFT calculations, why do the authors select an Fe₄-N₆ structure for Fe clusters? Please justify.
2. It is a convention to display XPS data with a reverse binding energy scale, i.e., with high binding energies to the left and low to the right. The authors may consider following this convention.
3. The Methods part should be a bit more detailed. What are the yields of CQD and the catalysts? How do the authors determine the concentration of the obtained CQD colloidal solution? How long was the catalyst ink used for the fuel cell testing stirred/sonicated? Who was the manufacturer of the carbon cloth used?
4. It would be better if the fuel cell stability of the catalyst is compared with the literature.
5. Please check the typography. The font of some abbreviations is obviously too large, such as P/P₀, P_{max}, E_{1/2}, ID/IG, etc.

Reviewer #2 (Remarks to the Author):

In this manuscript, Wan and coworkers reported a system involving Fe-N-C sites and N-coordinated Fe clusters for its application in acidic fuel cells. The physicochemical properties of catalysts were well investigated using HAADF-STEM, XPS, and XAS. DFT and MD simulations provided convincing support to the proposed reaction mechanism. Due to the strong electronic interplay within the short interaction distance, the binding strength of intermediates on SAs was optimized by surrounding clusters, which boosted the ORR reaction activity, as manifested by the half-wave potential of 0.81 V and the TOF of 2.82 s⁻¹. Moreover, the bond amplitude of Fe-N₄ was shortened by the incoherent vibration, thus leading to better demetallation resistance and stability in acidic conditions. Overall, it is an interesting work and offers some possible explanations about the existing problems of SA/cluster ORR catalysts,

especially on the stability enhancement. But at the same time, there are some key opinions and analyses presented in this manuscript that I could not fully agree with. Considering the impact of Nat. Commun., I suggest a major revision and another run of review. The following concerns should be well addressed, and I hope it can achieve a better quality before being considered.

1. Introduction. 1) Page 3, Line 9. The reasons for the inferior activity of M-N-C catalysts compared with Pt/C should be claimed. 2) Page 4, Line 2. Most metal nanoparticles/clusters are NOT “adsorbed onto the carbon support rather than fixed by chemical bonding” in the mentioned references. Fe-C is a strong chemical bonding, some of which can also act as a catalyst in an acidic medium. Authors should be careful and rigorous when overviewing research background.

2. In the acidic ORR performance tests (Fig. 3a), the FeSA, FeSA/cluster, and FeSA/particle display very similar Eonset and E1/2 values. The enhanced TOF value of FeSA/cluster actually stems from the lower Fe site density as determined by NO stripping. Speaking from the aspects of mass activity and atomic utilization efficiency, which are more important indices in practical fuel cells, the FeSA/cluster even exhibits much worse performance. How do the authors evaluate these disadvantages?

3. Carbon corrosion and demetallation are two factors that can affect each other. According to the literature report, *Angew. Chem. Int. Ed.*, 2015, 54, 12753-12757, carbon oxidation occurs at a high potential (> 0.9 V) with the destruction of active Fe-N-C sites which was believed to be the main reason for the activity degradation of catalysts in acidic medium. Since the authors observed negligible ID/IG ratio change in Raman spectra, the stable carbon structure should be responsible for the robust Fe-N-C sites and the enhanced stability. The sentence “the impact of carbon corrosion can be ruled out” on Page 13, Line 16 is not proper then. Except for the demetallation of SA, nanoparticles and clusters also suffer from the risk of leaching. In *ACS Energy Lett.*, 2019, 4, 1619 and *ACS Catal.* 2021, 11, 484, the low or non-active Fe clusters derived from Fe single atoms during PEMFC operation were confirmed to be easy to leach from the carbon matrix. What about the possibility of the aggregation of SA into clusters in this work? Is the bond in the Fe cluster stable enough? Will the formation and leaching of clusters affect the density of SA and weaken the synergies between SA and cluster?

4. The fabrications of FeSA, FeSA/cluster, and FeSA/particle catalysts were realized by varying the ratio of TPI precursors. By this approach, without any spatial confinement, the distribution and population of each Fe species are quite hard to control. For example, in Fig. 1d, the distance between SA and the adjacent cluster was measured at around 0.5 nm. However, there is a large amount of isolated SAs that distribute far away from the cluster and they are not taken into account. It is strongly suggested to reconsider the future of the proposed strategy of using Fe clusters as “boosters” for enhancing the intrinsic ORR activity/stability of M-N-C. The difficulty in the accurate synthesis of these structures would definitely cause reproducibility issues and increase the production cost. Besides, is it possible to precisely calculate the ratio of atomic Fe species and Fe clusters in the synthesized samples?

5. The authors can consider whether the computational model is consistent with the analysis results. 1) The average diameter of Fe clusters should be provided by a histogram. It looks like around 1 to 2 nm in Fig. 1d. In this case, clusters should consist of much more than four Fe atoms as shown in the DFT model. 2) In Fig. 2b, it was found that the FWHM and peak position of Fe²⁺ 2p_{3/2} signals in three samples are quite different, which makes the attribution of Fe²⁺ and Fe⁰ species questionable. Therefore, the conclusion “the positively charged iron species without zero-valent iron (Fe⁰) in FeSA/FeAC-2DNPC, indicating that Fe atoms in the clusters are possibly coordinated by the substrate

N/C atoms" on Page 8, Line 12 is not convincing. Even if these clusters are really anchored by N/C atoms, how can the authors decide it is a N atom but not a C atom? Please see Adv. Mater. 2020, 32, 2004900 and Small Methods 2021, 2001165. Will the theoretical results be different if the coordination atom changes?

6. Some missing data are suggested to be provided. 1) The ring and disk current curves recorded on RRDE. 2) The WT contour plots of Fe SA sample. 3) Fig. S18 only shows the LSV curves before and after 5k cycles, the chronoamperometric tests at 80 °C should also be given. 4) What is the spiking current shown in Fig. S17a?

Reviewer #3 (Remarks to the Author):

Comments:

This research has originality for synthesis of catalyst for efficient ORR system through the complex of Fe-N4 single atom site, Fe4-N6 nanocluster, Fe nanoparticle. Fe4-N= nanocluster can modulate the electronic structure of Fe-N4 site. Compared to Fe single atom catalysts, complex of Fe-N4 single atom site and Fe4-N6 nanocluster have high activity and durability. Since transition metal SACs can be replaced noble metal catalyst, attracting researchers in renewable energy society. In particular, Fe single atom catalyst has a problem of low durability in acid electrolytes, and this paper suggests a way to solve it. So, this paper is worth to be published in Nature Communications after revision of the manuscript following the comments below.

1) Through the Raman spectra data and difference of ORR limiting current density during the AST process, authors show that carbon corrosion does not occur. The pyrolysis temperature of M-N-C can affect the graphitic degree of carbon support and activity of the M-N4 site. Therefore, the authors need to show the relationship between the graphitic degree (Raman data) and activity (stability) difference at different temperature.

2) In DFT modeling, the authors suggested the complex of Fe-Nx site and Fe4-N6 site. In order to check the structure of the M-Nx site and the local coordination environment, fitting using EXAFS data is essential. However, the authors did not show the fitting data for the Fe-Nx site and Fe4-N6 site structure in EXAFS experimental data. The authors need to show experimental data for the Fe-Nx site and Fe4-N6 site structure presented in the modeling.

3) For site density and TOF measurements, the authors used nitrate stripping. Iron nanocluster and nanoparticles can be used for nitrate reduction (J. Water Process. Eng. 21,84-95,2018, Journal of Hazardous Materials,185,1513-1521, 2011). It is necessary to present reference or experimental results to ensure that the method using nitrate stripping is not affected by iron nanocluster and nanoparticles.

4) In Figure3a ORR performance data, FeSA/FeAC-2DNPC showed higher activity than FeSA/FeNP-2DNPC

and FeSA-2DNPC. In a previous paper, the synergistic effect of clusters and single atoms was shown in acid electrolytes (Small Methods, 5, 2001165, 2021). As in the previous paper, it is necessary to explain whether the cluster acts as an active site and produces a synergistic effect. In addition, FeSA/FeNP-2DNPC has a lower Tafel slope value and a higher TOF value than that of FeSA-2DNPC. It is necessary to present experimental or DFT modeling results to explain why FeSA/FeNP-2DNPC ORR activity is lower than that of FeSA-2DNPC in half-cell test.

5) In PEMFC full cell test, it should be clarified that the MEA is tested under atmospheric pressure (1 bar) or applied pressure (ambient pressure+1 bar=2 bar). It is not clear with the information provided in the experimental part.

Response to comments

Reviewer #1 (Remarks to the Author):

In the manuscript by Wan et al., the authors report the construction of Fe-N-C electrocatalyst with single-atom/cluster hybrid active sites for high-performance fuel cells. The iron single atoms and clusters are stabilized by N/C on carbon support and closely adjacent, thus creating a strong interaction. It is interesting that this strong interaction simultaneously increases the activity and stability of the single-atom active sites. In recent years, stability challenge has become a central issue for the development of M-N-C fuel cell catalysts, while the mitigation strategies are still limited. This work is the first to show that the synergy between single-atom/cluster could improve the fuel cell stability. The authors also performed molecular dynamics simulations to investigate the mechanism for the stability enhancement, and proposed a pinning effect of iron clusters, which can shorten the amplitude of Fe-N bonds of the adjacent Fe-N₄ active sites and thus improve their stability.

The presented data are well organized and the editing quality of the manuscript is acceptable. Due to the promising results and the apparent scientific quality, I would recommend the manuscript for publication in Nature Communications after solving the following issues.

R: We are grateful to the reviewer for the support and valuable comments/suggestions.

Comment 1: It is often very challenging to well define the structure of metal clusters. In DFT calculations, why do the authors select an Fe₄-N₆ structure for Fe clusters? Please justify.

R: We understand the reviewer's concern about the theoretic model. Our model is based on the electron microscopic characterization and literature. A survey of previous literature shows the following two types of practices when modeling the hybrid active site:

Type I (No. 1&2): the iron cluster is supported on pure carbon adjacent to an Fe-N_x site. However, we believe that Fe cluster is more likely to coordinate with N because N is ubiquitous and more electronegative than C.

Type II (No. 3&4): the iron cluster is connected to an Fe-N_x site via Fe-Fe bonding. This type is also less likely because the HADDF-STEM characterization shows that there is a gap

between the Fe single atom and the cluster.

Therefore, it is reasonable for us to setup a model of N-coordinated iron cluster adjacent to an Fe-N_x site. Here, Fe₄-N₆ is a simplified model for iron clusters as it is difficult to define the exact structure of metal clusters. In revision, we also try a larger cluster model consisting of 13 Fe atoms. The DFT calculation result shows that the regulation effect of Fe₁₃-N₆ is similar to that of Fe₄-N₆ (Supplementary Fig. 35).

No.	HADDF-STEM image	Structural model	Ref.
1		 Fe-N ₄ /Fe ₄ -C _x	Small Methods 5 , 2001165 (2021)
2		 Fe-N ₃ /Fe ₄ -C _x	Adv. Mater. , 2107291 (2021)
3		 Fe ₁ /Fe ₁₃ -Fe-N ₄	ACS Nano 13 , 11853–11862 (2019)
4		 Fe ₁₃ -Fe-N ₄	Angew. Chem. Int. Ed. , e202116068 (2021)
5		 Fe-N ₄ /Fe ₄ -N ₆	This work

Comment 2: It is a convention to display XPS data with a reverse binding energy scale, i.e., with high binding energies to the left and low to the right. The authors may consider following this convention.

R: According to the reviewer's suggestion, all XPS data have been revised with high binding energies to the left and low to the right.

Comment 3. The Methods part should be a bit more detailed. What are the yields of CQD and the catalysts? How do the authors determine the concentration of the obtained CQD colloidal solution? How long was the catalyst ink used for the fuel cell testing stirred/sonicated? Who was the manufacturer of the carbon cloth used?

R: We are sorry for the insufficient experimental details.

The yield of CQD was about 25 wt% from the raw ZIF-8 precursor. The yield of the catalyst was about 20 wt% based on CQD.

The concentration of the CQD colloidal solution was determined by a dry weighing method. A 100- μ l aliquot of CQD colloidal solution was pipetted onto microscope slides and dried on a hot plate, and then the remaining solid was weighed.

The ink was subjected to sonication for 10 min and stirring for 10 h to make a uniform suspension.

The carbon paper (GDS2240 with a microporous layer) was obtained from Ballard.

We have added the above information in the Methods section.

Comment 4: It would be better if the fuel cell stability of the catalyst is compared with the literature.

R: We appreciate the reviewer's valuable suggestion. The fuel cell stability of M-N-C catalysts measured under the 1 bar H₂-air is compared in Table R1.

The manuscript has been accordingly revised: "*This stability performance is very promising compared with reported M-N-C catalysts (Supplementary Table 7).*" Page 16.

Table R1. Comparison of PEMFC performance of Fe_{SA}/Fe_{AC}-2DNPC with other reported M-

N-C catalysts under 1 bar H₂-air.

Catalyst	Cell voltage (V)	Time (h)	Initial current density (mA cm ⁻²)	Final current density (mA cm ⁻²)	Decay rate (mA cm ⁻² h ⁻¹)	Decay rate (% h ⁻¹)	Ref.
Fe _{SA} /Fe _{AC} -2DNPC	0.5	150	440	365	0.5	0.11	This work
d-(Co _{NP} /Co _{SA} -N-C)	0.6	100	185	90	0.95	0.51	Energy Environ. Sci. 14 , 5958–5967 (2021)
P(AA-MA)(5-1)-Fe-N	0.55	40	81	73	0.2	0.25	Adv. Mater. 33 , 2006613 (2021)
Co(mIm)-NC(1.0)	0.7	100	125	100	0.25	0.2	Nat. Catal. 3 , 1044–1054 (2020)
FeN ₄ /HOPC-c-1000	0.55	100	638	395	2.43	0.38	Angew. Chem. Int. Ed. 59 , 2688–2694 (2020)
FeN _v /GM	0.4	93	449	253	2.1	0.47	Adv. Energy Mater. 9 , 1803737 (2019)
1.5Fe-ZIF	0.55	96	543	291	2.6	0.48	Energy Environ. Sci. 12 , 2548–2558 (2019)
TPI@Z8(SiO ₂)-650-C	0.5	20	365	222	7.15	1.95	Nat. Catal. 2 , 259–268 (2019)
20Mn-NC-second	0.7	120	28	17	0.09	0.33	Nat. Catal. 2 , 935–945 (2018)

20Co-NC-1100	0.7	100	40	15	0.25	0.63	Adv. Mater. 30 , 1706758 (2018)
Fe/N/CF	0.5*	100	350	209	1.41	0.4	Proc. Natl Acad. Sci. USA 112 , 10629–10634 (2015)
1/20/80-Z8- 1050 °C-15 min	0.5*	100	593	284	3.1	0.52	Nat. Commun. 2 , 416 (2011)
1/20/80-Z8- 1050 °C	0.5*	100	303	284	0.19	0.06	
PANI-FeCo-C(1)	0.4	700	347	337	0.014	0.004	Science 332 , 443–447 (2011)

*2 bar H₂–air

Comment 5: Please check the typography. The font of some abbreviations is obviously too large, such as P/P_0 , P_{\max} , $E_{1/2}$, I_D/I_G , etc.

R: We are sorry for the typos. We checked the manuscript thoroughly and corrected the errors in the revised manuscript.

Reviewer #2 (Remarks to the Author):

In this manuscript, Wan and coworkers reported a system involving Fe-N-C sites and N-coordinated Fe clusters for its application in acidic fuel cells. The physicochemical properties of catalysts were well investigated using HAADF-STEM, XPS, and XAS. DFT and MD simulations provided convincing support to the proposed reaction mechanism. Due to the strong electronic interplay within the short interaction distance, the binding strength of intermediates on SAs was optimized by surrounding clusters, which boosted the ORR reaction activity, as manifested by the half-wave potential of 0.81 V and the TOF of 2.82 s^{-1} . Moreover, the bond amplitude of Fe-N₄ was shortened by the incoherent vibration, thus leading to better demetallation resistance and stability in acidic conditions. Overall, it is an interesting work and offers some possible explanations about the existing problems of SA/cluster ORR catalysts, especially on the stability enhancement. But at the same time, there are some key opinions and analyses presented in this manuscript that I could not fully agree with. Considering the impact of Nat. Commun., I suggest a major revision and another run of review. The following concerns should be well addressed, and I hope it can achieve a better quality before being considered.

R: We are grateful to the reviewer for the support and valuable comments/suggestions.

Comment 1: Introduction. 1) Page 3, Line 9. The reasons for the inferior activity of M-N-C catalysts compared with Pt/C should be claimed. 2) Page 4, Line 2. Most metal nanoparticles/clusters are NOT “adsorbed onto the carbon support rather than fixed by chemical bonding” in the mentioned references. Fe-C is a strong chemical bonding, some of which can also act as a catalyst in an acidic medium. Authors should be careful and rigorous when overiewing research background.

R: We thank the reviewer’s valuable suggestion.

1) The main reasons for the inferior activity of M-N-C catalysts: i) the TOF of M-N_x is approximately an order of magnitude lower than those of Pt/C [*Nat. Mater.* **20**, 1385–1391 (2021); *Appl. Catal. B Environ.* **56**, 9–35 (2005)]; ii) insufficient accessible M-N_x active sites (<1 wt%) [*Nat. Catal.* **2**, 259–268 (2019)].

We have accordingly revised the manuscript: “*However, so far, the ORR activity of M–N–C in acidic media is still significantly lower than that of Pt-based catalysts due to insufficient accessible active sites and less competitive TOF.*” Page 3.

2) We are sorry for the inaccurate statement. We agree that the metal nanoparticles/clusters in the mentioned references might be connected to carbon support via Fe–C bonding. However, the dissolution of these clusters in acid indicates weak support-cluster interactions. We have accordingly modified this statement: “*It suggests that these metal NPs/ACs are weakly anchored (or bonded) on the carbon support, which may result in a very limited regulation effect on the electronic configuration of the SA sites.*” Page 4.

Comment 2: In the acidic ORR performance tests (Fig. 3a), the Fe_{SA}, Fe_{SA}/cluster, and Fe_{SA}/particle display very similar E_{onset} and E_{1/2} values. The enhanced TOF value of Fe_{SA}/cluster actually stems from the lower Fe site density as determined by NO stripping. Speaking from the aspects of mass activity and atomic utilization efficiency, which are more important indices in practical fuel cells, the Fe_{SA}/cluster even exhibits much worse performance. How do the authors evaluate these disadvantages?

R: The main advantage of Fe–N–C catalysts over Pt/C is negligible cost. For PGM catalysts, Pt mass activity and atomic utilization efficiency are important indices. But for PGM-free catalysts, it is more meaningful to evaluate the catalyst in terms of the area current density and volumetric current density of the fuel cell cathode. Therefore, US DOE targets are 0.44 A mg_{Pt}⁻¹@0.9V_{IR-free} for PGM catalysts, while 300 A cm⁻³@0.8V_{IR-free} and 0.044 A cm⁻²@0.9V_{IR-free} for PGM-free catalysts. In this context, Fe_{SA}/Fe_{AC}-2DNPC is obviously the best among three catalysts. Besides, the core value of this report is a new mechanism for the activity/stability enhancement of Fe–N–C catalysts by Fe clusters, whose cost is negligible.

Comment 3: Carbon corrosion and demetallation are two factors that can affect each other. According to the literature report, Angew. Chem. Int. Ed., 2015, 54, 12753-12757, carbon oxidation occurs at a high potential (> 0.9 V) with the destruction of active Fe-N-C sites which

was believed to be the main reason for the activity degradation of catalysts in acidic medium. Since the authors observed negligible I_D/I_G ratio change in Raman spectra, the stable carbon structure should be responsible for the robust Fe-N-C sites and the enhanced stability. The sentence “the impact of carbon corrosion can be ruled out” on Page 13, Line 16 is not proper then. Except for the demetallation of SA, nanoparticles and clusters also suffer from the risk of leaching. In ACS Energy Lett., 2019, 4, 1619 and ACS Catal. 2021, 11, 484, the low or non-active Fe clusters derived from Fe single atoms during PEMFC operation were confirmed to be easy to leach from the carbon matrix. What about the possibility of the aggregation of SA into clusters in this work? Is the bond in the Fe cluster stable enough? Will the formation and leaching of clusters affect the density of SA and weaken the synergies between SA and cluster?

R: We appreciate the reviewer’s insight into the degradation mechanisms. Specifically:

Carbon corrosion:

We fully agree that carbon oxidation is an indispensable cause of activity degradation. We apologize for inaccurate words that misled reviewers. Actually, the Raman spectra were not collected before and after the AST test. Instead, we compared the Raman spectra of the three fresh catalysts to check whether they differ in the degree of graphitization, as this is also a factor affecting catalyst stability. The results show that the three catalysts had a similar degree of graphitization. We have modified our statement: “*Therefore, the difference in the stability of the three catalysts is independent of the graphitic degree of the carbon supports.*” Page 13.

The possibility of the aggregation of SA into clusters:

According to the suggested literature, the formation of Fe clusters was observed after the load cycling at evaluated temperatures (80 °C) or a chronoamperometry test at the high constant potential of 0.85 V. Under other test conditions, the leached Fe ion would exit the catalyst layer [Angew. Chem. Int. Ed. **59**, 3235–3243 (2020)]. According to an operando (O_2) Mössbauer spectroscopy study [Nat. Catal. **4**, 10–19 (2020)], these clusters from Fe single-atoms presented as ferric oxides on Nafion ionomer without electrical connection with the cathode. Therefore, this kind of Fe clusters is totally different from the N/C-coordinated clusters in Fe_{SA}/Fe_{AC} -2DNPC.

In our demetalation experiment, the Fe loss in Fe_{SA}/Fe_{AC}-2DNPC after 5,000 potential cycles (0.6–1.0 V, 25 °C) was 15.5%. At 80 °C or in the chronoamperometric test at 0.75 V, the metal leaching should be severer. We do not exclude the possibility of the aggregation of leached Fe ions into Fe_xO clusters. However, the possible Fe_xO clusters are inactive toward ORR and do not interfere with the catalysis of Fe–N₄ sites.

The stability of the Fe cluster:

Our catalyst had been refluxed in a hot acid to remove soluble metal phases. Therefore, the remaining Fe–N₄ sites and Fe clusters should be stable in stagnant acidic conditions. However, under electrochemical reaction conditions, these iron species still suffered from demetalation as shown by the ICP results (Figure 3h). The Fe cluster is expected to be less stable than the Fe–N₄ site due to the presence of Fe–Fe bonds [*Chem* **5**, 2865–2878 (2019)]. Nevertheless, the HAADF-STEM image of the used Fe_{SA}/Fe_{AC}-2DNPC after 5k potential cycles (Supplementary Fig. 28) shows the well-retained single atoms and clusters, indicating the leaching rate was quite slow. We also noticed that the TOF of Fe_{SA}/Fe_{AC}-2DNPC decreased slightly from 2.60 to 2.39 s⁻¹, but still much higher than 1.79 s⁻¹ of the isolated Fe–N₄ site in Fe_{SA}-2DNPC, suggesting the interaction between SA and cluster still existed.

Effects of cluster formation and leaching:

As discussed above, the Fe ions leached from the Fe–N₄ site and then possibly formed Fe_xO clusters. In this case, the density of SA surely decreased as measured by our nitrite stripping experiment (18.7% reduction after 5k potential cycles), while the Fe_xO clusters (if any) were inactive and did not have a synergy with the Fe–N₄ sites. The pristine N/C-coordinated clusters also leached, which may weaken the synergies between SA and cluster to some extent, as manifested by the reduced TOF value.

Comment 4: The fabrications of Fe_{SA}, Fe_{SA}/cluster, and Fe_{SA}/particle catalysts were realized by varying the ratio of TPI precursors. By this approach, without any spatial confinement, the distribution and population of each Fe species are quite hard to control. For example, in Fig. 1d, the distance between SA and the adjacent cluster was measured at around 0.5 nm. However,

there is a large amount of isolated SAs that distribute far away from the cluster and they are not taken into account. It is strongly suggested to reconsider the future of the proposed strategy of using Fe clusters as "boosters" for enhancing the intrinsic ORR activity/stability of M-N-C. The difficulty in the accurate synthesis of these structures would definitely cause reproducibility issues and increase the production cost. Besides, is it possible to precisely calculate the ratio of atomic Fe species and Fe clusters in the synthesized samples?

R: We are fully aware of the imperfections of the current synthetic method, because it is difficult to precisely control the distribution and population of the iron species. Nevertheless, the key contribution of this report is a new enhancement strategy for the activity/stability of single-atom active sites. Following the reviewer's suggestion, we have reconsidered the future of this strategy and believe that improved synthetic approaches may give this strategy a promising future. For instance, a "precursor-preselection" strategy can guarantee the desired uniform composition of the cluster, as shown in our recent accurate synthesis of an Fe-Co double-atom catalyst [*Nano Res.* (2021). DOI: 10.1007/s12274-021-3966-y]. Using molecular precursors of multinuclear iron clusters, such as $\text{Fe}_4(\text{CO})_8(\text{pyridine})_4$, should be a future direction for fine control of the structure of the Fe cluster, which allows an accurate synthesis of Fe_1/Fe_n hybrid sites.

We had noticed there were many isolated SAs that distribute far away from the cluster, which should be responsible for the fast drop of fuel cell performance of $\text{Fe}_{\text{SA}}/\text{Fe}_{\text{NP}}\text{-2DNPC}$ during the initial 32 h (Fig. 4a).

As for the ratio of atomic Fe species and Fe clusters, it is hard to calculate via XPS or XAS because the Fe clusters are oxidized. A feasible method is to count a large number of single atoms and clusters in HAADF-STEM images. Based on three different areas of $\text{Fe}_{\text{SA}}/\text{Fe}_{\text{AC}}\text{-2DNPC}$ (Figure R1), a total of 908 iron single atoms and 90 clusters have been counted. Therefore, the ratio of atomic Fe species and Fe clusters in the catalyst is about 10:1. Some seemingly large clusters are composed of several small adjacent clusters while some bright spots may be due to the image overlap of single Fe atoms. We have added a description in the Catalyst synthesis and characterization section: "*From HAADF-STEM images, we estimate that*

the average diameter of Fe ACs is 0.7 nm and the ratio of SA to AC is about 10:1 (Supplementary Fig. 9). We note that there is a fraction of SAs far away from the ACs, which should behave like regular single-atom active sites.” Page 8.

Figure R1. Statistics of cluster size and SA-to-AC ratio in Fe_{SA}/Fe_{AC} -2DNPC. (a–c) Three different areas of HADDF-STEM images for statistics. The images are divided into small squares for counting single atoms. The number of single atoms in each square is marked in its upper left corner. (d) Size distribution histogram of the iron clusters in Fe_{SA}/Fe_{AC} -2DNPC.

Comment 5: The authors can consider whether the computational model is consistent with the analysis results. 1) The average diameter of Fe clusters should be provided by a histogram. It looks like around 1 to 2 nm in Fig. 1d. In this case, clusters should consist of much more than

four Fe atoms as shown in the DFT model. 2) In Fig. 2b, it was found that the FWHM and peak position of Fe^{2+} $2p_{3/2}$ signals in three samples are quite different, which makes the attribution of Fe^{2+} and Fe^0 species questionable. Therefore, the conclusion “the positively charged iron species without zero-valent iron (Fe^0) in $\text{Fe}_{\text{SA}}/\text{Fe}_{\text{AC}}\text{-2DNPC}$, indicating that Fe atoms in the clusters are possibly coordinated by the substrate N/C atoms” on Page 8, Line 12 is not convincing. Even if these clusters are really anchored by N/C atoms, how can the authors decide it is a N atom but not a C atom? Please see Adv. Mater. 2020, 32, 2004900 and Small Methods 2021, 2001165. Will the theoretical results be different if the coordination atom changes?

R: We thank the reviewer for the comments on the computational model.

1) As shown in Figure R1, the average diameter of Fe clusters is around 0.7 nm. It is a common practice in the literature to use a simplified model of four iron atoms to represent an iron cluster. We also tried a larger model consisting of 13 Fe atoms ($\text{Fe-N}_4\text{-OH}/\text{Fe}_{13}\text{-N}_6$) in DFT calculations. As shown in Figure R2, the boosting effect of Fe_{13} cluster is similar to that of the Fe_4 cluster and the limiting energy barrier is still much smaller than that of Fe-N_4 (0.38 vs. 0.53 eV).

Figure R2. Theoretical analysis of the activity of the hybrid active site with $\text{Fe}_{13}\text{-N}_6$ cluster.

(a) Model structure of $\text{Fe-N}_4/\text{Fe}_{13}\text{-N}_6$ used for theoretical calculation with a spontaneously

formed OH ligand. **(b)** Schematic ORR process on the Fe–N₄ site of Fe–N₄-OH/Fe₄-N₆. **(c)** Free energy diagram at 1.23 V for ORR over the Fe–N₄ site of Fe–N₄-OH/Fe₁₃-N₆. Free energy diagrams over the Fe–N₄ site of Fe–N₄-OH/Fe₄-N₆ and the bare Fe–N₄ site are shown for comparison.

2) We thank the reviewer for pointing out the inaccuracy in the fitting of XPS spectra. We checked our XPS raw data and found that we had mixed up the Fe 2p data for Fe_{SA}-2DNPC and Fe_{SA}/Fe_{NP}-2DNPC during the data fitting. We are very sorry for this mistake. The raw Fe 2p data are shown in Figure R3a. Although the signal-to-noise ratio is less satisfactory due to low iron contents, Fe⁰ (~706.7 eV) can be seen in Fe_{SA}/Fe_{NP}-2DNPC, while is not discernible in Fe_{SA}-2DNPC and Fe_{SA}/Fe_{AC}-2DNPC. According to the reviewer’s suggestion, we refit the Fe 2p spectra with the similar FWHM and the same peak position (Figure R3b). The manuscript has been accordingly revised: “Notably, the Fe 2p spectrum shows the positively charged iron species without obvious zero-valent iron (~706.7 eV) in Fe_{SA}/Fe_{AC}-2DNPC, indicating that Fe atoms in the clusters are possibly coordinated by the substrate N/C atoms.” Page 9.

Figure R3. (a) Raw Fe 2p XPS spectra of the catalysts. (b) Deconvoluted Fe 2p XPS spectra of the catalysts.

Since N is more electronegative than C, the Fe atom is more likely to coordinate with N,

which can be corroborated by the fact that Fe–N₄ sites are prevalent in Fe–N–C system while Fe–C₄ sites have been rarely reported. Therefore, it is reasonable for the Fe cluster to coordinate with N. We also calculated the carbon coordinated clusters (Fe–N₄/Fe₄–C₆), as presented in the recommended literature. As shown in Figure R4, the regulation effect of the Fe₄–C₆ cluster on the Fe–N₄ site is rather weak and limiting energy barrier is the same as the bare Fe–N₄ site.

The manuscript has been revised accordingly: “Two variants of the cluster are further investigated using models of Fe₁₃–N₆ and Fe₄–C₆. The calculations show that the N-coordinated iron cluster has a much more significant boosting effect on the adjacent Fe–N₄ site than the C-coordinated iron cluster, while the number of Fe atoms in the cluster plays a less significant role (Supplementary Figs. 35 and 36).” Page 19.

Figure R4. Theoretical analysis of the activity of the hybrid active site with Fe₄–C₆ cluster. (a) Model structure of Fe–N₄/Fe₄–C₆ used for theoretical calculation. (b) Schematic ORR process on the Fe–N₄ site of Fe–N₄/Fe₄–C₆. (c) Free energy diagram at 1.23 V for ORR over the Fe–N₄ site of Fe–N₄/Fe₄–C₆. Free energy diagram over the bare Fe–N₄ site is shown for comparison.

Comment 6: Some missing data are suggested to be provided. 1) The ring and disk current curves recorded on RRDE. 2) The WT contour plots of Fe SA sample. 3) Fig. S18 only shows the LSV curves before and after 5k cycles, the chronoamperometric tests at 80 °C should also be given. 4) What is the spiking current shown in Fig. S17a?

R: 1) The ring and disk current curves recorded on RRDE for Figure 3a,b are provided, as shown in Figure R5.

Figure R5. The ring and disk current curves recorded on RRDE for Figure 3a,b.

2) The WT contour plots of $\text{Fe}_{\text{SA}}\text{-2DNPC}$ and $\text{Fe}_{\text{SA}}/\text{Fe}_{\text{NP}}\text{-2DNPC}$ are provided, as shown in Figure R6. For $\text{Fe}_{\text{SA}}/\text{Fe}_{\text{NP}}\text{-2DNPC}$, the Fe–N/O scattering intensity maximum is not prominent due to the coverage of strong Fe–Fe scattering.

Figure R6. k^3 -weighted wavelet transforms of the experimental EXAFS spectra of $\text{Fe}_{\text{SA}}\text{-2DNPC}$, $\text{Fe}_{\text{SA}}/\text{Fe}_{\text{NP}}\text{-2DNPC}$, and references of Fe foil, FePc and Fe_2O_3 .

3) We have evaluated the stability of the catalysts at 80 °C by chronoamperometry at 0.75 V for 20 h as shown in Figure R7. We observed a 70% retention of the current density for Fe_{SA}/Fe_{AC}-2DNPC, outperforming Fe_{SA}-2DNPC (49%) and Fe_{SA}/Fe_{NP}-2DNPC (53%). The new data has been added in the revised manuscript: “*Chronoamperometry tests at 80 °C and 0.75 V for 20 h also show that Fe_{SA}/Fe_{AC}-2DNPC has the best current density retention (Supplementary Fig. 24).*” Page 13.

4) During the 20-h *i-t* test, we paused the test at the tenth hour and recorded LSV curves, then restarted the stability test for the remaining 10 hours. The spiking current indicated the lost current density can be partially recovered by the LSV potential cycling (0–1.1 V). This may be associated with the deactivation of active sites due to the oxidation of nearby carbon atoms. These oxygen groups could be partially removed by electrochemical reduction, leading to the recovery of activity to some extent [*Energy Environ. Sci.* **12**, 2548–2558 (2019)].

Figure R7. Stability tests by chronoamperometry at 80 °C. 20-h *i-t* tests at 0.75 V (a) for Fe_{SA}-2DNPC, Fe_{SA}/Fe_{AC}-2DNPC and Fe_{SA}/Fe_{NP}-2DNPC, during which the polarization curves were recorded initially and every ten hours (b,c,d). Test conditions: O₂-purged 0.5 M H₂SO₄, 300 rpm, 80 °C; catalyst loading of 0.4 mg cm⁻², graphite rod as counter electrode. LSV curves were recorded at 1,600 rpm.

Reviewer #3 (Remarks to the Author):

This research has originality for synthesis of catalyst for efficient ORR system through the complex of Fe-N₄ single atom site, Fe₄-N₆ nanocluster, Fe nanoparticle. Fe₄-N₆ nanocluster can modulate the electronic structure of Fe-N₄ site. Compared to Fe single atom catalysts, complex of Fe-N₄ single atom site and Fe₄-N₆ nanocluster have high activity and durability. Since transition metal SACs can replace noble metal catalyst, attracting researchers in renewable energy society. In particular, Fe single atom catalyst has a problem of low durability in acid electrolytes, and this paper suggests a way to solve it. So, this paper is worth to be published in Nature Communications after revision of the manuscript following the comments below.

R: We are grateful to the reviewer's support and valuable comments/suggestions.

Comment 1: Through the Raman spectra data and difference of ORR limiting current density during the AST process, authors show that carbon corrosion does not occur. The pyrolysis temperature of M-N-C can affect the graphitic degree of carbon support and activity of the M-N₄ site. Therefore, the authors need to show the relationship between the graphitic degree (Raman data) and activity (stability) difference at different temperature.

R: We appreciate the suggestion. We adjusted the temperature of the second pyrolysis (T_2 /°C) from 800 to 1000 °C. The samples were denoted as $x\%-T_2$, where $x\%$ means the mass ratio of TPI relative to CQD in the precursor. The Raman spectra of these samples are shown in Figure R8. The ORR activity of the samples was evaluated by rotating ring disk electrode (RRDE) in O₂-saturated 0.5 M H₂SO₄ solution at room temperature. The stability was evaluated by 10,000 potential cycling from 0.6 to 1.0 V vs. RHE in O₂-purged 0.5 M H₂SO₄ at a scan rate of 50 mV s⁻¹ and a rotation rate of 300 rpm at room temperature. The results are shown in Figure R9. Based on these data, we correlate the graphitic degree (I_D/I_G), activity ($E_{1/2}$), and stability ($\Delta E_{1/2}$ after 10,000 cycles) with T_2 (Figure R10). We can draw the following conclusions:

1) Increasing T_2 can increase the graphitic degree of the catalyst, while the Fe content does not influence the graphitic degree of the catalyst.

2) Increasing T_2 can increase the catalyst activity, possibly due to the thermally driven

evolution of ideal local structures of Fe–N₄ sites, with one exception (30%TPI-1000). We speculate that the high Fe content plus high pyrolysis temperature led to Fe agglomeration into inactive Fe nanoparticles and thus decreased the activity.

3) Increasing T_2 can improve the catalyst stability, which could be partially attributed to the enhanced graphitic degree. We note that when T_2 was increased from 900 to 1000 °C, the stability of the 15%- T_2 series showed a greater improvement relative to the other two groups, suggesting the emergence of a new stability enhancement mechanism.

The manuscript has been accordingly revised:

“The pyrolysis temperature was optimized to 1000 °C to achieve the best activity and stability (Supplementary Figs. 5–7) The high temperature is crucial for the formation of optimal active sites and highly graphitic carbon support.” Page 7.

“The approximately equal I_D/I_G ratios indicate a similar degree of graphitization of the three catalysts, which is controlled by the pyrolysis temperature regardless of the iron content (Supplementary Fig. 7a).” Page 13.

Figure R8. Raman spectra of the catalysts. The temperature of the second pyrolysis (T_2 /°C) was adjusted from 800 to 1000 °C. The samples were denoted as $x\%-T_2$, where $x\%$ means the mass ratio of TPI relative to CQD in the precursor.

Figure R9. ORR polarization curves of $x\%$ - T_2 before and after 10,000 potential cycles (0.6–1.0 V vs. RHE) in O_2 -purged 0.5 M H_2SO_4 at room temperature.

Figure R10. Correlations between a) graphitic degree (I_D/I_G), b) ORR activity ($E_{1/2}$) and c) stability ($\Delta E_{1/2}$) of the catalysts with second pyrolysis temperature (T_2).

Comment 2: In DFT modeling, the authors suggested the complex of $Fe-N_x$ site and Fe_4-N_6 site. In order to check the structure of the $M-N_x$ site and the local coordination environment,

fitting using EXAFS data is essential. However, the authors did not show the fitting data for the Fe-N_x site and Fe₄-N₆ site structure in EXAFS experimental data. The authors need to show experimental data for the Fe-N_x site and Fe₄-N₆ site structure presented in the modeling.

R: We thank the reviewer's valuable suggestion. The FT-EXAFS spectrum of Fe_{SA}/Fe_{AC}-2DNPC was well fitted in *R* space using backscattering paths of Fe-N/O and Fe-Fe (two Fe-Fe paths with different distances). The fitting results are shown in Figure R11 and Table R2. The coordination numbers of Fe-N/O and Fe-Fe were about 5.17 and 0.72, respectively. These numbers are reasonable as the XAS signal is an average probing of the single-atom sites and clusters in the catalyst. Based on the SA (Fe-N₄-O₂) to AC (Fe₄-N₆) ratio of 10:1, we can calculate that the coordination numbers of Fe-N/O and Fe-Fe are about 4.71 and 0.43, which are close to the fitting results.

The manuscript has been accordingly revised: “*The FT-EXAFS spectrum was well fitted using backscattering paths of Fe-N/O and Fe-Fe (Fig. 2d, Supplementary Table 1). The coordination numbers of Fe-N/O and Fe-Fe were about 5.17 and 0.72, respectively.*” Page 9.

Figure R11. Fourier-transformed experimental EXAFS spectrum of Fe_{SA}/Fe_{AC}-2DNPC and its fitting spectrum.

Table R2. Fitting results of Fe K-edge EXAFS spectrum of Fe_{SA}/Fe_{AC}-2DNPC.

Sample	Path	N	R (Å)	σ^2 (Å ²)	ΔE_0 (eV)	S ₀ ²	R factor (%)
Fe _{SA} /Fe _{AC} - 2DNPC	Fe–N/O	5.17	2.02	0.011	3.6	0.9	0.49
	Fe–Fe	0.38	2.55	0.006			
	Fe–Fe	0.33	3.05	0.006			

N is coordination number, *R* is the distance between absorber and backscatter atoms, σ^2 is Debye-Waller factor to account for both thermal and structural disorders, ΔE_0 is inner potential correction; *R* factor indicates the goodness of the fit. Error bounds (accuracies) that characterize the structural parameters obtained by EXAFS spectroscopy were estimated as $N \pm 20\%$; $R \pm 1\%$; $\sigma^2 \pm 20\%$; $\Delta E_0 \pm 20\%$. *S*₀² was fixed to 0.9 as determined from Fe foil fitting. Fitting range: $2.5 \leq k$ (1/Å) ≤ 10.8 and $1 \leq R$ (Å) ≤ 3 .

Comment 3: For site density and TOF measurements, the authors used nitrate stripping. Iron nanocluster and nanoparticles can be used for nitrate reduction (J. Water Process. Eng. 21, 84-95, 2018, Journal of Hazardous Materials, 185, 1513-1521, 2011). It is necessary to present reference or experimental results to ensure that the method using nitrate stripping is not affected by iron nanocluster and nanoparticles.

R: Thanks for the reminding. Actually, we determined catalyst site density by using electrochemical **nitrite** (NO₂[−]) stripping rather than **nitrate** (NO₃[−]) stripping, according to the method presented by Kucernak et al. [*Nat. Commun.* 7, 13285 (2016)]. This process is illustrated below:

The nitrite adsorption was conducted by dipping RDE into 125 mM NaNO₂ solution for 5 minutes at open circuit potential. According to the study by FUJIE et al. [*Water Res.* 35, 2789 – 2793 (2001)], nitrite could indeed be reduced by metallic iron. However, the reduction rate of nitrite was rather low at pH = 5 (15 mM/h). Thus, it can be estimated that during the nitrite

adsorption process, the concentration of NaNO_2 solution does not decrease by more than 1%, not to mention that the metallic iron in the catalysts is minimal. More importantly, the major products are ammonium (NH_4^+) and nitrogen gas (N_2), which should not interfere with adsorption and subsequent stripping of nitrite on the active Fe-N_x sites. It is also noticed that in the original report by Kucernak et al, a bimetallic iron-cobalt catalyst (containing metallic iron) had also been tested and they concluded that the method should work provided the catalyst is not highly active for hydrogen evolution, which would mask the stripping charge. Therefore, the stripping method is generally applicable for the vast majority of Fe-N-C catalysts.

We have added a brief note in the Methods section to clarify this issue: “*We note that nitrite ions may be reduced by the trace amount of metallic iron (if any) in the catalysts. However, the major products are ammonium and nitrogen gas, which should not interfere with adsorption and subsequent stripping of nitrite on Fe-N_x sites.*” Page 27.

Comment 4: In Figure 3a ORR performance data, $\text{Fe}_{\text{SA}}/\text{Fe}_{\text{AC}}\text{-2DNPC}$ showed higher activity than $\text{Fe}_{\text{SA}}/\text{Fe}_{\text{NP}}\text{-2DNPC}$ and $\text{Fe}_{\text{SA}}\text{-2DNPC}$. In a previous paper, the synergistic effect of clusters and single atoms was shown in acid electrolytes (Small Methods, 5, 2001165, 2021). As in the previous paper, it is necessary to explain whether the cluster acts as an active site and produces a synergistic effect. In addition, $\text{Fe}_{\text{SA}}/\text{Fe}_{\text{NP}}\text{-2DNPC}$ has a lower Tafel slope value and a higher TOF value than that of $\text{Fe}_{\text{SA}}\text{-2DNPC}$. It is necessary to present experimental or DFT modeling results to explain why $\text{Fe}_{\text{SA}}/\text{Fe}_{\text{NP}}\text{-2DNPC}$ ORR activity is lower than that of $\text{Fe}_{\text{SA}}\text{-2DNPC}$ in half-cell test.

R: We investigated the ORR activity of the cluster by DFT calculations. The Fe_4 surface in $\text{Fe-N}_4\text{-OH}/\text{Fe}_4\text{-N}_6$ is chosen as the active site. As shown in Figure R12, a large energy uphill (1.37 eV) for OH^* desorption is observed, implying the formation of a permanent OH ligand. After the attachment of the OH ligand, however, the situation becomes even worse (Figure R13). The ultralarge ΔG (3.64 eV) for the formation of OH^* ($\text{O}^* + \text{H}^+ + e^- \rightarrow \text{OH}^*$) indicates that the reaction is impossible to proceed. Therefore, the iron cluster, at least in the considered model, does not act as an active site. Instead, it acts as a booster to enhance the intrinsic ORR

activity/stability of the Fe–N₄ site.

The manuscript has been accordingly revised: “*The Fe₄ in Fe–N₄-OH/Fe₄-N₆ is predicted with inferior activity (Supplementary Figs. 33 and 34), indicating the cluster mainly acts as an activity booster.*” Page 19.

Figure R12. Theoretical analysis of the activity of the iron cluster (Fe₄-N₆) in the hybrid active site. (a) Model structure of Fe–N₄-OH/Fe₄-N₆ used for theoretical calculation. (b) Schematic ORR process on the Fe₄ site of Fe–N₄-OH/Fe₄-N₆. (c) Free energy diagram at 1.23 V for ORR over the Fe₄ site of Fe–N₄-OH/Fe₄-N₆.

Figure R13. Theoretical analysis of the activity of the OH-modified iron cluster (Fe₄-N₆-OH)

in the hybrid active site. (a) Model structure of Fe–N₄-OH/Fe₄-N₆-OH used for theoretical calculation. (b) Schematic ORR process on the Fe₄ site of Fe–N₄-OH/Fe₄-N₆-OH. (c) Free energy diagram at 1.23 V for ORR over the Fe₄ site of Fe–N₄-OH/Fe₄-N₆-OH.

On the activity difference between Fe_{SA}/Fe_{NP}-2DNPC and Fe_{SA}-2DNPC:

Actually, Fe_{SA}/Fe_{NP}-2DNPC has a slightly lower Tafel slope value and a slightly higher TOF value (57.2 mV dec⁻¹, 2.01 s⁻¹) than those of Fe_{SA}-2DNPC (57.8 mV dec⁻¹, 1.79 s⁻¹). The apparent ORR activity (expressed by $E_{1/2}$) of Fe_{SA}/Fe_{NP}-2DNPC (0.786 V) is slightly lower than that of Fe_{SA}-2DNPC (0.795 V). As you see, the electrochemical properties of these two catalysts are rather similar. This is because the iron nanoparticles in Fe_{SA}/Fe_{NP}-2DNPC are encapsulated by thick layers of graphitic carbon (typically > 2 nm, as shown in Supplementary Fig. 13). The possible synergistic effect between single-atom Fe–N₄ and the Fe NPs should be rather weak. Therefore, the TOF of the active sites in Fe_{SA}/Fe_{NP}-2DNPC is only marginally enhanced. In our study, we conclude that the closely adjacent iron clusters serve as a more powerful promoter of the activity of Fe–N₄ compared with the encapsulated iron NPs. On the other hand, the formation of iron NPs consumes considerable iron sources and leads to the decreased density of Fe–N₄ sites, which is confirmed by the nitrite stripping experiments. Fe_{SA}/Fe_{NP}-2DNPC has an SD of 30.2 μmol g⁻¹, which is much smaller than that of Fe_{SA}-2DNPC (41.4 μmol g⁻¹). The slightly high TOF but much lower SD make the apparent activity of Fe_{SA}/Fe_{NP}-2DNPC lower than that of Fe_{SA}-2DNPC in half-cell test.

Comment 5: In PEMFC full cell test, it should be clarified that the MEA is tested under atmospheric pressure (1 bar) or applied pressure (ambient pressure+1 bar=2 bar). It is not clear with the information provided in the experimental part.

R: We are sorry for the insufficient experimental information. The pressure conditions for each fuel cell data were specified in the figure captions. For example, 1 bar H₂-O₂ means that the absolute pressure for H₂ and O₂ is both 1 bar, which is achieved by applying 0.5 bar backpressure. For clarity, we have added this information to the Methods section.

REVIEWERS' COMMENTS

Reviewer #1 (Remarks to the Author):

All of the comments have been well addressed and the current form can be accepted.

Reviewer #2 (Remarks to the Author):

All my concerns were resolved in the revised manuscript, and I think it is now suitable for publication in Nat. Commun.

Reviewer #3 (Remarks to the Author):

The comments by the referees are well-addressed in the revised version of this paper. It is of great importance to develop highly stable M-N-C catalysts. With unique idea, the authors successfully developed highly stable M-N-C catalysts. This concept can be extended to the field where the M-N-C catalyst is required. I would recommend the publication of this work in Nature Communications without any further revision.